# Going Further: Flatness at the Rescue of Early Stopping for Adversarial Example Transferability

## Abstract

Transferability is the property of adversarial examples to be misclassified by other models than the surrogate model for which they were crafted. Previous research has shown that early stopping the training of the surrogate model substantially increases transferability. A common hypothesis to explain this is that deep neural networks (DNNs) first learn robust features, which are more generic, thus a better surrogate. Then, at later epochs, DNNs learn non-robust features, which are more brittle, hence worst surrogate. We demonstrate that the reasons why early stopping improves transferability lie in the side effects it has on the learning dynamics of the model. We first show that early stopping benefits the transferability of non-robust features. Then, we establish links between transferability and the exploration of the loss landscape in the parameter space, on which early stopping has an inherent effect. More precisely, we observe that transferability peaks when the learning rate decays, which is also the time at which the sharpness of the loss significantly drops. This leads us to evaluate the training of surrogate models with seven minimizers that minimize both loss value and loss sharpness. One of such optimizers, SAM always improves over early stopping (by up to 28.8 percentage points). We also uncover that the strong regularization induced by SAM with large flat neighborhoods is tightly linked to transferability. Finally, the best sharpness-aware minimizers are competitive with other training techniques, and complementary to other types of transferability techniques.

## 1 Introduction

State-of-the-art Deep Neural Networks (DNNs) are vulnerable to imperceptible worst-case inputs perturbations, so-called adversarial examples (Biggio et al., 2013; Szegedy et al., 2013). These perturbations are not simple flukes of specific representations because some are simultaneously adversarial against several independently trained models with distinct architectures (Goodfellow et al., 2014). This observation leads to the discovery of the *transferability* of adversarial examples, i.e., an adversarial example against a model is likely to be adversarial against another model. This phenomenon is not well understood but has practical implications. Indeed, practitioners cannot rely on security by obscurity. Attackers can apply white-box attacks to their *surrogate model* to fool an unknown *target model*. These types of attack are called transfer-based back-box attacks. They do not require *any* query access to the model to craft adversarial examples. Crafting highly transferable adversarial examples for distinct architectures is still an open problem (Naseer et al., 2022) and an active area of research (Benz et al., 2021; Dong et al., 2018; Gubri et al., 2022a;b; Li et al., 2018; Lin et al., 2019; Springer et al., 2021; Wu et al., 2020; Xie et al., 2019; Zhao et al., 2022). Understanding the underlining characteristics that drive transferability provides insights into how DNNs learn generic representations.

Despite strong interest in transferability, little attention has been paid to how to train better surrogate models. The most commonly used method is arguably *early stopping* (Benz et al., 2021; Zhang et al., 2021; Nitin, 2021) – which is originally a practice to improve natural generalization and avoid overfitting. The commonly accepted hypothesis to explain why early stopping improves transferability is that an early stopped DNN is composed of more robust features, whereas the fully trained counterpart has more brittle non-robust features (Benz et al., 2021; Zhang et al., 2021; Nitin, 2021).

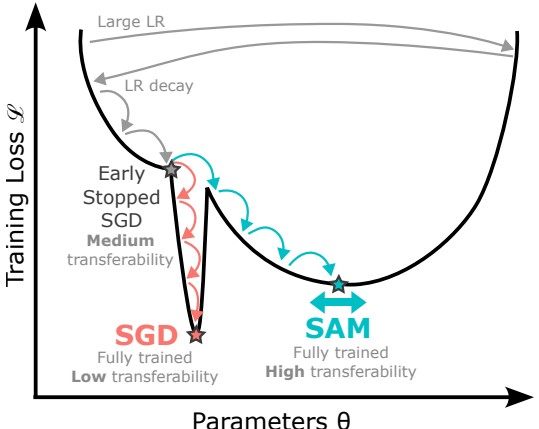

Figure 1: Illustration of the relation between the training dynamics of the surrogate model, sharpness, and transferability. Before the learning rate decays, training tends to "cross the valley" with plateauing transferability. A few iterations after the decay of the learning rate, early stopped SGD achieves its best transferability (gray). In the following epochs, SGD falls progressively into deep, sharp holes in the parameter space with poor transferability (red). l-SAM (blue) avoids these holes by minimizing the maximum loss around an unusually large neighborhood (thick blue arrow).

In this paper, we invalidate this hypothesis empirically and uncover other explanations behind the effectiveness of early stopping, and more generally on how to achieve better surrogate training. We observe in Section 3 that early stopping also improves transferability from and to models composed of *non-robust* features. We formulate an alternative hypothesis that the success of early stopping is closely related to the dynamics of the exploration of the loss surface. Section 4 establishes that transferability peaks a few iterations of SGD after the decay of the learning rate while the loss sharpness in the weight space drops. Later, the transferability slowly decreases and the sharpness slowly increases. Based on these observed correlations, we show in Section 5 that flat-minima optimizers significantly increase the transferability of a surrogate model by minimizing its sharpness. In particular, we reveal that the stronger regularization induced by Sharpness-Aware Minimizer (SAM) with unusually large neighborhood (l-SAM), improves transferability specifically, since l-SAM and SGD have a similar natural generalization. We conclude that this strong regularization alters the exploration of the loss landscape by avoiding deep, sharp holes where the learned representation is too specific. Finally, in Section 6 we evaluate l-SAM and two variants competitively against other training procedures and complementarily to other categories of transferability techniques.

Figure 1 illustrates the insights and grounded principles to improve transferability that our contribution brings:

- The learning rate decay allows the exploration of the loss landscape to go down the valley. After a few iterations, SGD reaches its best transferability ("early stopped SGD", gray star). The sharpness is temporarily contained.

- As training with SGD continues, sharpness increases and transferability decreases. The fully trained model (red star) is a suboptimal surrogate. SGD falls into deep, sharp holes where the representation is too specific.

- SAM explicitly minimizes sharpness and avoids undesirable holes. Transferability is maximum after a full training (blue star) when SAM is applied over a large neighborhood (l-SAM, thick blue arrow).

## 2 RELATED WORK

**Transferability techniques.** The transferability of adversarial examples is a prolific research topic (Benz et al., 2021; Dong et al., 2018; Gubri et al., 2022a;b; Li et al., 2018; Lin et al., 2019; Springer et al., 2021; Wu et al., 2020; Xie et al., 2019; Zhao et al., 2022). Zhao et al. (2022) recently categorize

transferability techniques and recommend evaluating techniques by comparing them against each other within each category. Section 5 follows this recommendation. Gradient-based transferability techniques can be decomposed into model augmentation, data augmentation, attack optimizers, and feature-based attacks. In Section 5, we show that our method improves the following techniques when combined. Model augmentation adds randomness to the weights or the architecture to avoid specific adversarial examples: GN (Li et al., 2018) uses dropout or skip erosion, SGM (Wu et al., 2020) favors gradients from skip connections during the backward pass, LGV (Gubri et al., 2022b) collects models along the SGD trajectory during a few additional epochs with a high learning rate. Data augmentation techniques transform the inputs during the attack: DI (Xie et al., 2019) randomly resizes the input, SI (Lin et al., 2019) rescales the input, and VT (Wang & He, 2021) smooths the gradients locally. Attack optimizers smooth updates during gradient ascent with momentum (MI, Dong et al. (2018)) or Nesterov accelerated gradient (NI, Lin et al. (2019)).

**Training surrogate models.** Despite the important amount of work on transferability, the way to train an effective single surrogate base model has received little attention in the literature (Zhao et al., 2022). Benz et al. (2021); Nitin (2021); Zhang et al. (2021) point that early stopping SGD improves transferability. Springer et al. (2021) propose SAT, slight adversarial training that uses tiny perturbations to filter out some non-robust features. Section 5 evaluates SAT. Our approach sheds new light on the relation between flatness and transferability. Springer et al. (2021) implicitly flatten the surrogate model, since adversarial trained models are flatter than their naturally trained counterparts (Stutz et al., 2021). We observe a similar implicit link with early stopping in Section 4. Gubri et al. (2022b) propose the surrogate-target misalignment hypothesis to explain why flat minima in the weight space are better surrogate models. We show that LGV, their model augmentation technique, is complementary to ours.

**Early stopping for transferability.** Several works (Benz et al., 2021; Zhang et al., 2021; Nitin, 2021) point out that fully trained surrogate models are not optimal for transferability. To explain this observation, they propose a hypothesis based on *the perspective of robust and non-robust features (RFs/NRFs)* from Ilyas et al. (2019). Ilyas et al. (2019) disentangles features that are highly predictive and robust to adversarial perturbations (RFs), and features that are also highly predictive but non-robust to adversarial perturbations (NRFs). According to Benz et al. (2021); Nitin (2021), the training of DNNs mainly learns RFs first and then learns NRFs. NRFs are transferable (Ilyas et al., 2019), but also brittle. RFs in a tiny input neighborhood, called *slightly RFs*, improve transferability (Zhang et al., 2021; Springer et al., 2021): the input neighborhood is sufficiently small for an attack to find adversarial examples in a larger radius, and slightly RFs are less brittle than NRFs. Models at earlier epochs would be composed of more slightly RFs, thus being better surrogate models. Section 3 provides some observations that tend to refute this hypothesis. Instead, Sections 4 and 5 suggest that the success of early stopping is correlated with the training dynamics and sharpness.

**Sharpness and natural generalization** Several training techniques increase natural generalization and reduce loss sharpness in the weight space. SWA (Izmailov et al., 2018) averages the weights at the last epochs to find a flatter solution. SAM (Foret et al., 2020) minimizes the maximum loss around a neighborhood by performing a gradient ascent step followed by a gradient descent step. At the cost of one additional forward-backward pass per iteration, SAM avoids deep, sharp holes on the surface of the loss landscape (Kaddour et al., 2022). Several variants exist that improve natural generalization (Kwon et al., 2021; Zhuang et al., 2022) or efficiency (Liu et al., 2022; Du et al., 2021). Nevertheless, the relationship between sharpness and natural generalization is subject to scientific controversy (Andriushchenko et al., 2023; Wen et al., 2023). In Sections 5 and 6, we explore the use of SWA, SAM and six variants to train better surrogate models.

## 3 ANOTHER LOOK AT THE NON-ROBUST FEATURES HYPOTHESIS ABOUT EARLY STOPPING

In this section, we point the flaws of the robust and non-robust features (RFs/NRFs) hypothesis Benz et al. (2021); Zhang et al. (2021); Nitin (2021) to explain the success of early stopping for transferability. According to this hypothesis, earlier representations are more transferable than their fully trained counterparts, because they contain more slightly RFs than NRFs. Slightly RFs are

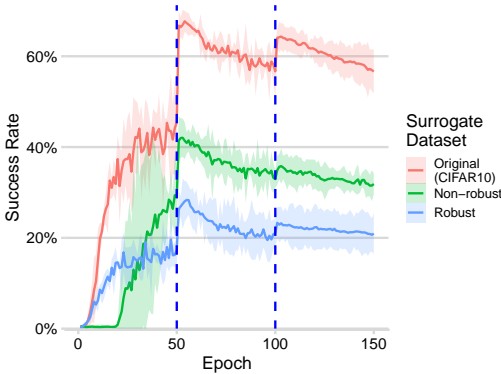 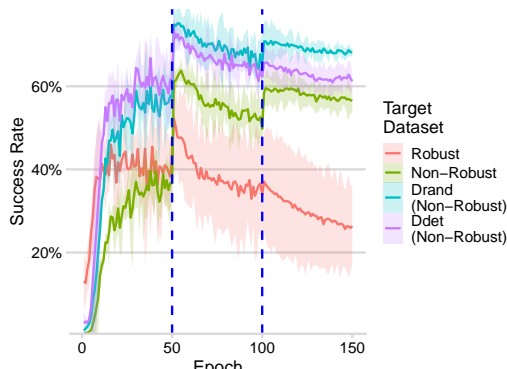

Figure 2: Early stopping improves the transferability *from* surrogate models trained on both robust and non-robust datasets. Average success rate evaluated over ten target models trained on the original CIFAR-10 dataset, from a ResNet-50 surrogate model trained for a number of epochs (x-axis) on the datasets $D_R$ (blue) and $D_{NR}$ (green) of Ilyas et al. (2019) modified from CIFAR-10 (red). We craft all adversarial examples from the same subset of the original CIFAR-10 test set. Average (line) and confidence interval of $\pm$ two standard deviations (colored area) of three training runs. Appendix C contains the details per target.

Figure 3: Early stopping improves the transferability *to* target models trained on both robust and non-robust datasets. Success rate from a ResNet-50 trained for a number of epochs (x-axis) on the original CIFAR-10 dataset, to ResNet-50 targets trained on the robust dataset $D_R$ (red), and the three non-robust datasets $D_{NR}$ (green), $D_{rand}$ (blue) and $D_{det}$ (purple) of Ilyas et al. (2019) modified from CIFAR-10. The perturbation norm $\varepsilon$ is $16/255$ for the $D_R$ target, $2/255$ for the $D_{NR}$ target and $1/255$ for the $D_{rand}$ and $D_{det}$ targets to adapt to the vulnerability of target models (the order of lines cannot be compared). Average (line) and confidence interval of $\pm$ two standard deviations (colored area) of three training runs. Best seen in colors.

features that are robust to tiny worst-case perturbations, and NRFs are features that are not. See Section 2 for more details.

**Early stopping indeed increases transferability.** First, we check that a fully trained surrogate model is not optimal for transferability. We train two ResNet-50 surrogate models on CIFAR-10 and ImageNet using standard settings. Appendix B reports the success rates on CIFAR-10 and ImageNet of the BIM attack applied at every epoch and evaluated on 10 fully trained target models per dataset. For both datasets and diverse targeted architectures, the optimal epoch for transferability occurs around one or two thirds of training[1]. Indeed, early stopping increases transferability.

**Early stopping improves transferability from both surrogates trained on robust and non-robust features.** We show that early stopping works *similarly* well on surrogate models trained on robust and non-robust datasets. We retrieve the robust and non-robust datasets from Ilyas et al. (2019), that are altered from CIFAR-10 to mostly contain RFs and, respectively, NRFs. We train two ResNet-50 models on both datasets with SGD (hyperparameters reported in Appendices B and C). Figure 2 shows the transferability across training epochs, averaged over the ten regularly trained targets. The success rates of both robust and non-robust surrogate models evolve similarly (scaled by factor) to the model trained on the original dataset: transferability peaks around the epochs 50 and 100 and decreases during the following epochs. This observation is valid for all ten targets (details in Appendix C). According to the RFs/NRFs hypothesis, we expected "X-shaped" transferability curves: increasing transferability from NRFs and strictly decreasing transferability from RFs (after initial convergence). The RFs/NRFs hypothesis does not describe why early learned NRFs are better for transferability than fully learned NRFs.

---

[1] Transferability decreases along epochs, except for the two vision transformers targets on ImageNet where the transferability is stable at the end of training.

**Early stopping improves transferability to both targets trained on robust and non-robust features.** We observe that an early stopped surrogate model trained on the original dataset is best to target both targets composed of RFs and NRFs. Here, we keep the original CIFAR-10 dataset to train the surrogate model. We target four ResNet-50 models trained on the robust and non-robust datasets of Ilyas et al. (2019)[2]. Figure 3 shows that the same epoch of standard training is optimal for attacking all four models, i.e., composed of either RFs or NRFs. The RFs/NRFs hypothesis fails to explain why early stopping is best to target NRFs.

Overall, we provide new evidence that early stopping for transferability acts similarly on robust and non-robust features. We do not observe an inherent trade-off between RFs and NRFs. Since the higher the transferability, the more similar the representations are, we conclude that the early trained representations are more similar to both RFs and NRFs than their fully trained counterparts. Therefore, the hypothesis that early stopping favors RFs over NRFs does not hold. We conjecture that a phenomenon orthogonal to RFs/NRFs explains why fully trained surrogates are not optimal.

## 4 STOPPING EARLIER: TRANSFERABILITY AND TRAINING DYNAMICS

This section explores the relationship between the training dynamics of the surrogate model and its transferability. In particular, we observe that following the learning rate step decays, transferability peaks when sharpness drops.

**Transferability peaks when the LR decays.** We point out the key role of the LR decay in the success of early stopping for transferability. The optimal number of surrogate training epochs for transferability occurs a couple of epochs after the decay of the LR. We train a ResNet-50 surrogate model for 150 epochs on CIFAR-10, using the standard LR schedule of Engstrom et al. (2019) which divides the LR by 10 at epochs 50 and 100. For the ten targets considered individually, the highest transferability is between epochs 51 and 55 (Appendix B). Figure 4 shows that transferability suddenly peaks after both LR decays (red line). We train on ImageNet a ResNet-50 surrogate model for 90 epochs with LR decay at epochs 30 and 60. The highest transferability per target occurs either after the first decay (epochs 31 or 35) or after the second one (epochs 62 or 67), except for both vision transformer targets, where transferability plateaus at a low success rate after the second decay. Overall, the success of early stopping appears to be related to the exploration of the loss landscape, which is governed by the learning rate.

**Consistency of the peak of transferability across training.** The peak of transferability described above can be consistently observed at any point of training (after initial convergence). Here, we modify the standard double decay LR schedule to perform a single decay at a specified epoch. The learning rate is constant (0.1) until the specified epoch, where it is ten times lower for the rest of the training. We evaluate the transferability of five surrogates with a decay at, respectively, epoch 25, 50, 75, 100 and 125. In Figure 4, we observe a similar transferability peak for all these surrogates, except for the decay at epoch 25 where the decay occurs before the end of the initial convergence (details per target in Appendix D). The consistency of the peak of transferability across training epochs is valid for all individual targets. We add as baseline the constant learning rate (at 0.1). Without LR decay, transferability plateaus after initial convergence. Therefore, we conclude that the step decay of the LR enables early stopping to improve transferability.

**Sharpness drops when the LR decays.** When the LR decays, the sharpness in the parameter space drops. We compute two sharpness metrics at every epoch using the PyHessian library (Yao et al., 2019) on a random subset of a thousand examples from the CIFAR-10 train dataset. The largest eigenvalue of the Hessian measures the sharpness of the sharpest direction in the weight space (worst-case sharpness), and the trace of the Hessian measures the total sharpness of all directions in weight space (average sharpness). Figure 5 reproduces the largest Hessian eigenvalue (red) and the Hessian trace (blue) per the training epoch of our standard CIFAR-10 surrogate. We observe that both types of sharpness decrease abruptly and significantly immediately after both LR decays at epochs 50 and 100. Simultaneously, transferability peaks (orange).

---

[2]In this experiment, we include two additional non-robust datasets $D_{\text{rand}}$ and $D_{\text{det}}$ from Ilyas et al. (2019). By construction, their *only* useful features for classification are NRFs. We did not include them in the previous experiment because training on them is too unstable.

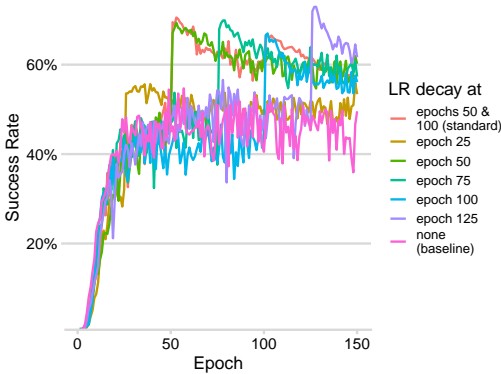
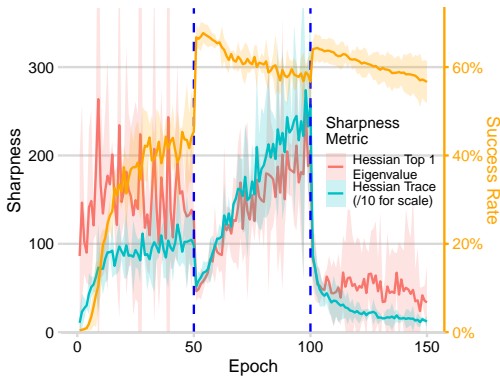

Figure 4: Transferability peaks when the learning rate decays at any epochs. Average success rate evaluated over ten target models from a ResNet-50 surrogate model trained for a number of epochs (x-axis) on CIFAR-10. The learning rate is divided by 10 once during training at the epoch corresponding to the color. Red is our standard schedule, with two decays at epochs 50 and 100. Pink is the baseline of constant learning rate. Best seen in colors.

Figure 5: Sharpness drops when the learning rate decays. Largest eigenvalue of the Hessian (red) and trace of the Hessian (blue) for all training epochs (x-axis) on CIFAR-10. Average success rate on ten targets (orange, right axis). Average (line) and confidence interval of ± two standard deviations (colored area) of three training runs. Vertical bars indicate the learning rate step decays. Best seen in colors.

We conclude that *the effect of early stopping on transferability is tightly related to the dynamics of the exploration of the loss surface*, governed by the learning rate. Overall, Figure 1 illustrates our observations:

1. Before the LR decays, the training bounces back and forth crossing the valley from above (top gray arrows). See Appendix D for an extended discussion on the matter.

2. After the LR decays, training goes down the valley. Soon after, SGD has its best transferability ("early stopped SGD" gray star). Sharpness is reduced.

3. When learning continues, the training loss decreases and sharpness slowly increases. SGD finds a "deep hole" of the loss landscape, corresponding to specific representations that have poor transferability ("fully trained SGD" red star).

## 5    GOING FURTHER: FLATNESS AT THE RESCUE OF SGD

Since transferability peaks to its higher value when sharpness drops, in this section, we explore how to improve transferability by minimizing the sharpness of the surrogate model. First, we show that seven training techniques that minimize both the loss value and the loss sharpness can train better surrogate models. Second, we uncover that SAM (and five variants) with unusually large flat neighborhoods induces a stronger regularization that specifically increases transferability.

**Minimizing sharpness improves transferability.** The training techniques known to decrease the sharpness of the models train better surrogate representations. We evaluate the transferability of seven training techniques belonging to two families, SWA and SAM (see Section 2). SWA (Izmailov et al., 2018) decreases sharpness implicitly by averaging the weights collected by SGD. Our SWA surrogate is the average of the weights obtained by our standard SGD surrogate at the end of the last 25% epochs[3]. Figure 6 shows that SWA (yellow) improves the success rate compared to fully trained SGD (red) on both datasets. On ImageNet, SWA beats the early stopped SGD surrogate, but not on CIFAR-10. Indeed, SWA helps to find flatter solutions than those found by SGD, but SWA is confined to the same basin of attraction (Kaddour et al., 2022). To remediate to this

---

[3]We also update the batch-normalization statistics of the SWA model with one forward pass over the training data on CIFAR-10 (10% on ImageNet).

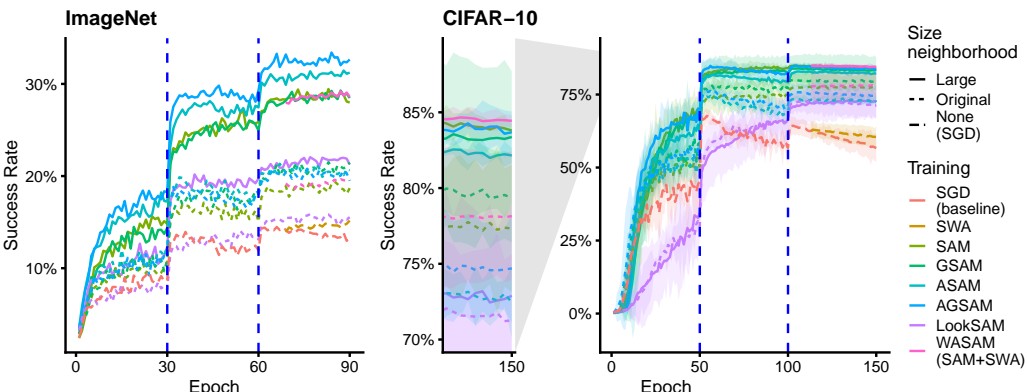

Figure 6: SAM variants and SWA improve transferability over SGD, and SAM with large neighborhoods over the original SAM. Average success rate evaluated over ten target models from a ResNet-18 surrogate model trained for a number of epochs (x-axis) on ImageNet (left), and from a ResNet-50 on CIFAR-10 (right). SAM and its variants are trained with both the original size of flat neighborhood (dotted, $\rho = 0.05$ except $\rho = 0.5$ for adaptive variants) and the larger size that we tuned for transferability (plain). Red is our standard SGD surrogate. Best seen in colors.

issue, we also train several surrogate models with SAM (Foret et al., 2020) and its variants, i.e., GSAM (Zhuang et al., 2022), ASAM (Kwon et al., 2021), AGSAM (GSAM+ASAM), WASAM (SAM+SWA, Kaddour et al. (2022)), and LookSAM (Liu et al., 2022). SAM explicitly minimizes sharpness during training by solving a min-max optimization problem. At each iteration step, SAM first maximizes the loss in a neighborhood to compute a second gradient that is used to minimize the loss (see Appendix E for an illustration). We train one model per SAM variant using the original SAM hyperparameter ($\rho = 0.05$). Figure 6 shows that SAM and its variants (dotted lines) train surrogate models that have a significantly higher transferability than fully trained SGD, early stopped SGD and SWA, on both datasets. On ImageNet, the success rate of SAM averages over the ten targets at 18.7%, compared to 13.3% for full training with SGD, 14.5% for SGD at its best (epoch 66) and 15.2% for SWA, and respectively 77.3%, 56.6%, 67.7% (epoch 54) and 60.5% on CIFAR-10. SAM finds different basins of attractions than SGD (Foret et al., 2020; Kaddour et al., 2022). Therefore, some basins of attraction are better surrogate than others, and explicitly minimizing sharpness reaches better ones.

**Strong regularization from large flat neighborhoods significantly improves transferability.** We uncover that the size of the flat neighborhood of SAM and its variants induces a regularization that is tightly linked to transferability. We observe that SAM and its variants with uncommonly large flat neighborhoods train significantly and consistently better surrogate models. SAM seeks neighborhoods with uniformly low loss of size controlled by its $\rho$ hyperparameter. We tune it on CIFAR-10 on distinct validation sets of natural examples, target, and surrogate models (details in Appendix E). For all SAM variants, the optimal $\rho$ for transferability is always larger than the original $\rho$, and unusually large compared to the range of values used for natural accuracy. Indeed, we find $\rho$ of 0.3 optimal for SAM, and Foret et al. (2020) originally uses $\rho$ of 0.05. Kaddour et al. (2022) and Zhuang et al. (2022) tune $\rho$ with a maximum of, respectively, 0.2 and 0.3. Figure 6 reports the transferability of SAM and its variants with both the original $\rho$ (dotted) and with the larger $\rho$ found optimal on CIFAR-10 (plain). All SAM variants train a better surrogate model with large $\rho$ values[4]. In the following, we denote **l-SAM** for SAM with large $\rho$ (0.3), and similarly **l-AGSAM** and **l-LookSAM** (respectively, 4[5] and 0.3). Kaddour et al. (2022) show that changing $\rho$ ends up in

---

[4]LookSAM shows a slower learning behavior over the epoch, compared to other SAM variants. LookSAM is a more efficient variant that computes the additional SAM gradient only once each five optimizer iteration. Therefore, this behavior is expected.

[5]l-AGSAM uses $\rho$ value of 4, since as suggested by Kwon et al. (2021) adaptive variants should use $\rho$ 10 times larger. We found our observations consistent with this recommendation.

Table 1: Success rate and computational cost of surrogate training techniques on ImageNet and CIFAR-10. The success rate is averaged on ten targets from a ResNet-50 surrogate with a maximum perturbation $L_\infty$ norm $\varepsilon$ of $4/255$ (other norms in Appendix). The computational overhead is computed from the number of forward-backward passes compared to SGD. Bold is best. In %.

|  | Success Rate | | Computation Cost | |
| --- | --- | --- | --- | --- |
| Surrogate | ImageNet | CIFAR-10 | ImageNet | CIFAR-10 |
| Fully Trained SGD | 17.81 | 56.06 | $\times\,1$ | $\times\,1$ |
| Early Stopped SGD | 19.97 | 70.16 | $\times\,0.77$ | $\times\,0.36$ |
| SAT (Springer et al., 2021) | 49.74 | 62.45 | $\times\,4$ | $\times\,8$ |
| SWA | 20.83 | 60.26 | $\times\,1.00$ | $\times\,1.00$ |
| l-SAM (ours) | 48.75 | 85.50 | $\times\,2$ | $\times\,2$ |
| l-AGSAM (ours) | **53.14** | **85.72** | $\times\,2$ | $\times\,2$ |
| l-LookSAM (ours) | 33.17 | 77.49 | $\times\,1.23$ | $\times\,1.22$ |

different basins of attraction. Therefore, the stronger regularization induced by l-SAM avoids large sharp holes on top of the loss surface, and significantly improves transferability.

**The benefits of the strong regularization from large flat neighborhoods are specific to transferability.** The stronger regularization of SAM with a large value of $\rho$ is specifically related to transferability. First, this strength of regularization may degrade natural accuracy. On ImageNet with ResNet-18, the top-1 accuracy of SAM with large $\rho$ is equal to 67.89%, less than SAM with the original $\rho$ (70.29%) and even less than fully trained SGD (69.84%). This observation extends to ResNet-50 and to the other variants of SAM on ImageNet (see Appendix E). Therefore, the improvement in generalization of adversarial examples cannot be explained by an improvement in natural generalization (better fit to the data). Second, unlike SAM, a stronger regularization of weight decay decreases transferability, showing a specific relation between transferability and SAM. We train multiple surrogate models using SGD with different values of weight decay. The optimal weight decay value for the ResNet-50 surrogate is the same value used to train the target model (see Appendix F for details). Therefore, not all regularization schemes help to train a better surrogate model.

Overall, we show that the sharpness of the surrogate model is tightly related to transferability:

- Minimizing implicitly or explicitly the loss sharpness trains better surrogate models.
- The strong regularization induced by SAM with large $\rho$ avoids deep sharp minima in favor of unusually large flat neighborhoods that contain more generic representations.
- The stronger SAM regularization is tailored for transferability: it can reduce natural accuracy, and other strong regularization schemes, such as weight decay, do not aid in transferability.

## 6 PUTTING IT ALL TOGETHER: IMPROVING TRANSFERABILITY TECHNIQUES WITH SHARPNESS MINIMIZATION

In this section, we show that explicitly minimizing sharpness is a competitive technique for training surrogate models and complements well other transferability techniques. To benchmark our principle against related work, we adhere to the best practices suggested by Zhao et al. (2022). Specifically, we evaluate the benefits of minimizing sharpness on large neighborhoods against other surrogate training techniques (same category), and also assess their complementarity with techniques from different categories. All our code and models are available on GitHub[6].

**Minimizing sharpness improves over competitive techniques.** l-SAM and l-AGSAM are competitive alternatives to existing surrogate training techniques, and l-LookSAM offers good transferability for a small computational overhead. For a fair comparison, we choose the epoch of the

---

[6]URL redacted for double-blind review.

Table 2: Success rate of other categories of transferability techniques applied on the standard SGD base surrogate and on our l-SAM base surrogate. The success rate is averaged on our ten ImageNet targets from ResNet-50 models with a maximum perturbation $L_\infty$ norm $\varepsilon$. Bold is best. In %.

| Attack | $\varepsilon = 2/255$ | | $\varepsilon = 4/255$ | | $\varepsilon = 8/255$ | |
|---|---|---|---|---|---|---|
| | SGD | l-SAM | SGD | l-SAM | SGD | l-SAM |
| **Model Augmentation Techniques** | | | | | | |
| GN (Li et al., 2018) | 12.9 | **28.8** | 27.8 | **52.8** | 46.5 | **71.0** |
| SGM (Wu et al., 2020) | 11.7 | **24.3** | 29.3 | **51.5** | 55.6 | **76.2** |
| LGV (Gubri et al., 2022b) | 24.8 | **25.2** | 53.5 | **54.7** | 72.1 | **73.7** |
| **Data Augmentation Techniques** | | | | | | |
| DI (Xie et al., 2019) | 22.1 | **42.0** | 47.0 | **72.5** | 69.4 | **86.9** |
| SI (Lin et al., 2019) | 10.8 | **28.8** | 26.9 | **56.7** | 49.9 | **77.2** |
| VT (Wang & He, 2021) | 10.5 | **31.9** | 24.9 | **59.4** | 43.0 | **78.5** |
| **Attack Optimizers** | | | | | | |
| MI (Dong et al., 2018) | 12.3 | **32.0** | 26.8 | **59.6** | 46.3 | **78.3** |
| NI (Lin et al., 2019) | 8.3 | **20.6** | 22.3 | **46.5** | 43.9 | **70.5** |

early stopped SGD surrogate by evaluating a validation transferability at every training epoch[7]. We retrieve the SAT (Slight Adversarial Training) ImageNet pretrained model used by Springer et al. (2021), and we train SAT on CIFAR-10 using their adversarial training hyperparameters. Table 1 reports the average success rate of the aforementioned techniques, alongside their computational overhead. This overhead is quantified as the ratio of forward-backward passes needed to train the surrogate model to those required for training with SGD. On both datasets, l-AGSAM is the best surrogate. l-AGSAM beats the transferability of SAT, while dividing the training cost by two on ImageNet and four on CIFAR-10. Nevertheless, l-AGSAM doubles the computational number of forward-backward passes compared to SGD. By computing the additional SAM gradient only once per five iterations, l-LookSAM is a viable alternative to contain the computational overhead to 1.23, while having higher transferability than SGD. Overall, sharpness-aware minimizers with large flat neighborhoods offer a good trade-off between transferability and computation.

**Minimizing sharpness trains better base models for complementary techniques.** l-SAM is a good base model to combine with existing model augmentation, data augmentation, and attack optimization transferability techniques. These categories aim complementary objectives: model and data augmentations reduce the tendency of the attack to overfit the base model by adding randomness to gradients. Attack optimizers intend to smooth the gradient updates. Table 2 reports the success rate of eight transferability techniques combined with our l-SAM base model on ImageNet. For all adversarial perturbation norms $\varepsilon$, l-SAM provides a base model that improves every eight techniques, compared to the standard fully trained SGD surrogate, from 0.4 to 35.5 percentage points.

## 7 CONCLUSION

Overall, our insights into the behavior of SGD through the lens of transferability drive us to a successful approach to train better surrogate models with limited computational overhead. Our observations lead us to reject the hypothesis that early stopping benefits transferability due to an inherent trade-off between robust and non-robust features. Instead, we explain the success of early stopping in relation to the dynamics of the exploration of the loss landscape. After the learning rate decays, SGD drives down the valley and progressively falls into deep, sharp holes. These fully trained representations are too specific to generate highly transferable adversarial examples. We remediate this issue by explicitly minimizing sharpness in unusually large neighborhoods. Avoiding those large sharp holes proves to be useful in improving transferability on its own and in complement with existing transferability techniques.

---

[7]To ensure no data leakage that could violate our no-query threat model, we craft one thousand adversarial examples from images of a validation set and evaluate them against a distinct set of target models.

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

# APPENDIX

These supplementary materials contain the following sections:

- Appendix A details the experimental settings,
- Appendix B reports the transferability and the natural accuracy by epochs of surrogate trained with SGD on CIFAR-10 and ImageNet,
- Appendix C reports additional results of Section 3 "Another Look at the Non-Robust Features Hypothesis about Early Stopping",
- Appendix D reports additional results of Section 4 "Stopping Earlier: Transferability and Training Dynamics",
- Appendix E reports additional results of Section 5 "Going Further: Flatness At The Rescue of SGD",
- Appendix F reports the results about transferability with respect to the weight decay of the surrogate,
- Appendix G reports additional results of Section 6 "Putting It All Together: Improving Transferability Techniques With Sharpness Minimization".

## A  EXPERIMENTAL SETTINGS

This section describes the experimental settings used in this article. The experimental setup is standard for transfer-based attacks.

- Our source code used to train and evaluate models is publicly available on GitHub at this URL: **redacted for double-blind review**.
- Our trained models on both CIFAR-10 and ImageNet are publicly distributed through HuggingFace at this URL: **redacted for double-blind review**.

**Target models.**   All our target models on CIFAR-10 are fully trained for 150 epochs with SGD using the hyperparameters reported in Table 3. For a fair comparison, *the baseline surrogate is trained with SGD using the same hyperparameters as the targets*. On CIFAR-10, we target the following ten architectures: ResNet-50 (the surrogate with the same architecture is an independently trained model), ResNet-18, ResNet-101, DenseNet-161, DenseNet-201, WideResNet-28-10, VGG13, VGG19, Inception v3 and ConvMixer. On ImageNet, the target models are the pretrained models distributed by PyTorch. The ten target architectures on ImageNet are the following: ResNet-50, ResNet-152, ResNeXt-50 32X4D, WideResNet-50-2, DenseNet-201, VGG19, GoogLeNet (Inception v1), Inception v3, ViT B 16 and Swin S. Additionally, we train a "validation" set of architectures on CIFAR-10 to select hyperparameters independently of reported results. This set is composed of: ResNet-50 (another independently trained model), ResNet-34, ResNet-152, DenseNet-121, DenseNet-169, WideResNet-16-8, VGG11, VGG16, GoogLeNet (Inception v1) and MLPMixer. This validation set of target models on ImageNet is composed of the following architectures: ResNet-50 (another independently trained model), ResNet-101, ResNeXt-101 64X4D, WideResNet101-2, VGG16, DenseNet121, ViT B 32 and Swin B.

**Surrogate models trained with SGD.**   We train the surrogate models on CIFAR-10 and ImageNet using SGD with the standard hyperparameters of the robustness library (Engstrom et al., 2019) (Table 3). Due to computational limitations on ImageNet, we limit the number of epochs to 90, reusing the same hyperparameters as Ashukha et al. (2020).

**Surrogate models trained with SAM and its variants.**   We train surrogate models with SAM using the same hyperparameters as the models trained with SGD for both datasets. We integrate the SAM optimizer into the robustness library Engstrom et al. (2019). The unique hyperparameter of SAM is $\rho$, which is set to $0.05$ as the original paper for both datasets for the original SAM surrogate. The l-SAM surrogate is trained with SAM with $\rho$ equal to $0.4$. The $\rho$ values used to train the variants of SAM are reported in Table 3. We use official or popular implementations of ASAM, GSAM,

AGSAM and LookSAM, following the original paper. LookSAM is an efficient variant of SAM that computes the additional gradient of SAM only once per five training iterations. As reported by Liu et al. (2022), LookSAM is unstable at the beginning of training. Liu et al. (2022) solve this issue using a learning rate with warmup. Since we wanted to use the same learning rate schedule for all training techniques, we added another type of warmup. LookSAM computes the additional SAM gradient at all training iterations during the first three epochs. Our LookSAM is equivalent to SAM before the fourth epoch. This simple solution is enough for LookSAM to converge. This computational overhead is taken into account in the computational cost reported in Table 1.

**Surrogate models of competitive techniques (Section 6).** To compare with competitive training techniques on ImageNet, we retrieve the original models of SAT Springer et al. (2021), an adversarially trained model with a small maximum $L_2$ norm perturbation $\varepsilon$ of 0.1 and with the PGD attack applied with 3 steps and a step size equal to $2\varepsilon/3$. On CIFAR-10, we reuse the best hyperparameters of Springer et al. (2021) to adversarially train the SAT surrogate model with a maximum $L_2$ norm $\varepsilon$ of 0.025 and PGD with 7 steps and a step size of $0.3\varepsilon$. For a fair comparison, we choose the best checkpoint of the early stopped SGD surrogate by evaluating the transferability of every training epoch. For each epoch, we craft 1,000 adversarial examples from a distinct validation set of original examples and compute their success rate over a distinct set of validation target architectures. On CIFAR-10, the selected epoch is 54, and 66 on ImageNet. All the other hyperparameters not mentioned in this paragraph are the same as those used to train the surrogates with SGD.

**Attack.** Unless specified otherwise, we use the BIM (Basic Iterative Method, equivalently called I-FGSM) Kurakin et al. (2017) which is the standard attack for transferability Benz et al. (2021); Dong et al. (2018); Gubri et al. (2022a;b); Li et al. (2018); Lin et al. (2019); Springer et al. (2021); Wu et al. (2020); Xie et al. (2019); Zhao et al. (2022). By default, the maximum $L_\infty$ perturbation norm $\varepsilon$ is set to $4/255$. We use the BIM hyperparameters tuned by Gubri et al. (2022a;b) on a distinct set of validation target models: BIM performs 50 iterations with a step size equal to $\varepsilon/10$. Unless specified otherwise, we craft adversarial examples from a subset of 1,000 natural test examples that are correctly predicted by all target models. We repeat the experiments on CIFAR-10 three times, each run with a different random seed, an independently sampled subset of original examples, and *an independently trained surrogate model*. For every CIFAR-10 experiment, we train three times each surrogate model to estimate correctly the randomness of an attacker training a surrogate model to perform an attack. The success rate is the misclassification rate of these adversarial examples evaluated on one target model. We report the average success rate across the three random seeds, along with a confidence interval of plus/minus two times the empirical standard deviation.

**Threat model.** We study the threat model of untargeted adversarial examples: the adversary's goal is misclassification. We consider the standard adversary capability for transfer-based black-box attacks, where the adversary does not have query access to the target model. Query-based attacks are another distinct family of attacks.

**Implementation.** The source code for each experiment is available on GitHub. Our models are distributed through HuggingFace. We use the torchattacks library (Kim, 2020) to craft adversarial examples with the BIM attacks and four transferability techniques, namely LGV, DI, SI, VT, MI and NI. We reuse the original implementations of GN and SGM to "patch" the surrogate architecture, and use the TorchAttacks implementation of BIM on top. The software versions are the following: Python 3.10.8, PyTorch 1.12.1, Torchvision 0.13.1, and TorchAttacks 3.3.0.

**Infrastructure.** For all experiments, we use Tesla V100-DGXS-32GB GPUs on a server with 256GB of RAM, CUDA 11.4, and the Ubuntu operating system.

Table 3: Hyperparameters used to train surrogate models.

| Training | Hyperparameter | Dataset | Value |
|---|---|---|---|
| All | Number of epochs | CIFAR-10 | 150 |
| | | ImageNet | 90 |
| | Initial learning rate | All | 0.1 |
| | Learning rate decay | CIFAR-10 | Step-wise /10 each 50 epochs |
| | | ImageNet | Step-wise /10 each 30 epochs |
| | Momentum | All | 0.9 |
| | Batch-size | CIFAR-10 | 128 |
| | | ImageNet | 256 |
| | Weight decay | CIFAR-10 | 0.0005 |
| | | ImageNet | 0.0001 |
| SAM | $\rho$ | All | 0.05 for SAM, 0.4 for l-SAM |
| GSAM | $\rho$ | All | 0.05 for GSAM, 0.2 for l-GSAM |
| | $\alpha$ | All | 0.15 |
| LookSAM | $\rho$ | All | 0.05 for LookSAM, 0.3 for l-LookSAM |
| | $k$ | All | 5 |
| | SAM Warmup | All | 3 epochs |
| ASAM | $\rho$ | All | 0.5 for ASAM, 3 for l-ASAM |
| AGSAM | $\rho$ | All | 0.5 for AGSAM, 4 for l-AGSAM |
| | $\alpha$ | All | 0.15 |

# B TRANSFERABILITY AND NATURAL ACCURACY BY EPOCHS

Early stopping clearly benefits transferability for all ten targets on CIFAR-10 and all ten targets on ImageNet (except for the two Vision Transformers, where the transferability plateaus). We reproduce below the success rates for all target models from the ResNet-50 surrogate model on both CIFAR-10 (Figure 7) and ImageNet (Figure 9) datasets. We also report the evolution of the natural accuracy for both CIFAR-10 (Figure 8) and ImageNet (Figure 10).

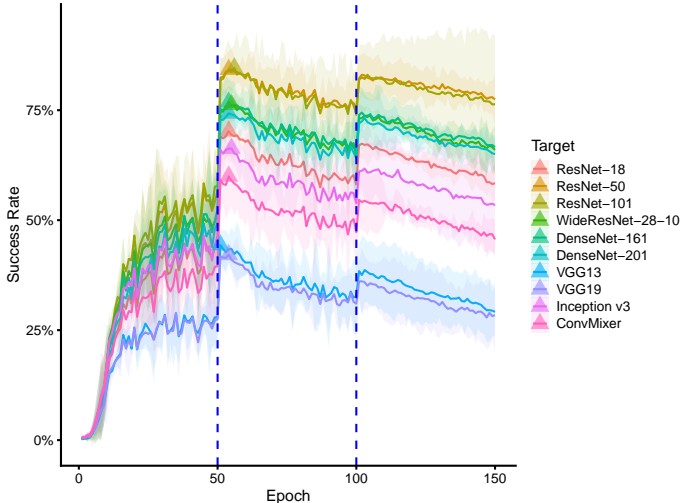

Figure 7: Early stopping improves transferability consistently across target models on CIFAR-10. Success rate evaluated on ten target models (color) from a ResNet-50 surrogate model trained for a number of epochs (x-axis) on the CIFAR-10 dataset. We report the average over three random seeds (line) and the confidence interval of two standard deviations (colored area). Vertical bars indicate the step decays of the learning rate. Triangles indicate the epochs corresponding to the highest success rate per target.

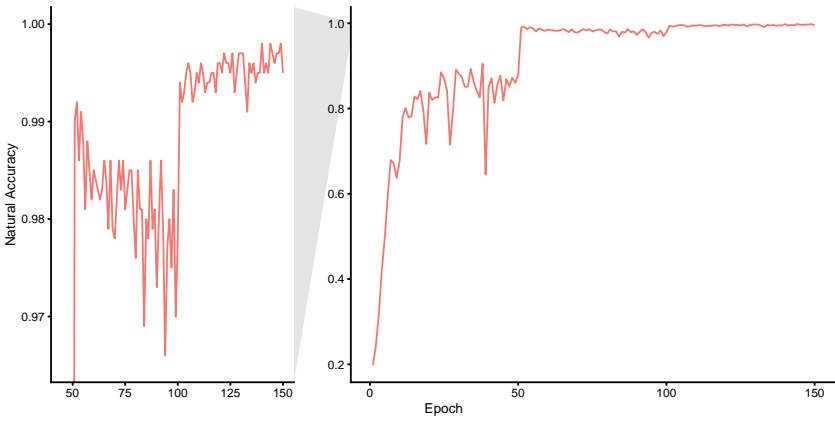

Figure 8: The natural accuracy increases at the end of training, whereas transferability decreases. Natural test accuracy of the ResNet-50 surrogate model trained for a number of epochs (x-axis) on the CIFAR-10 dataset. Evaluated on the test subset used to craft adversarial examples in Figure 7.

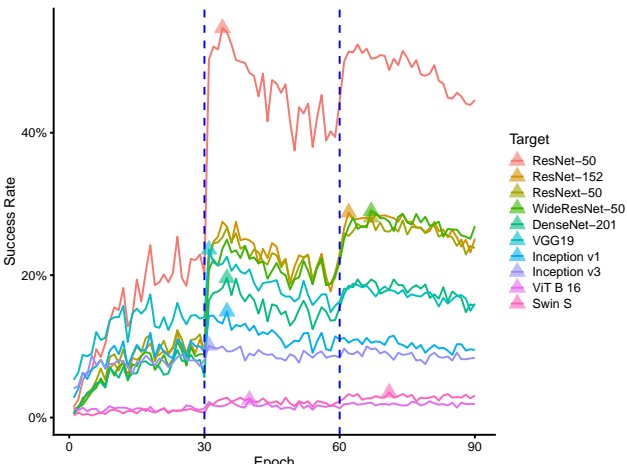

Figure 9: Early stopping improves transferability to various target models on ImageNet, except to vision transformers (ViT-B-16 and Swin-S) against which the success rate plateaus at the end of training. Success rate evaluated on ten target models (colour) from a ResNet-50 surrogate model trained for a number of epochs (x-axis) on the ImageNet dataset. Vertical bars indicate the step decays of the learning rate. Triangles indicate the epochs corresponding to the highest success rate per target.

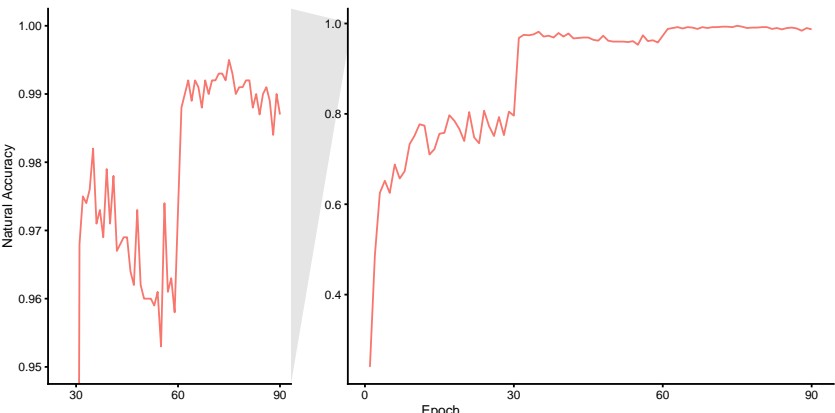

Figure 10: Natural test accuracy of the ResNet-50 surrogate model trained for a number of epochs (x-axis) on the ImageNet dataset. Evaluated on the test subset used to craft adversarial examples in Figure 9.

# C    ANOTHER LOOK AT THE NON-ROBUST FEATURES HYPOTHESIS ABOUT EARLY STOPPING

This section contains detailed results of Section 3. Figure 11 reports the transferability per target of the experiment that shows the success of early stopping for surrogates trained on both robust and non-robust datasets. For this experiment, we divided by two the initial learning rate (0.05) when training on $D_{\text{NR}}$ due to instabilities during training when trained with a learning rate of 0.1.

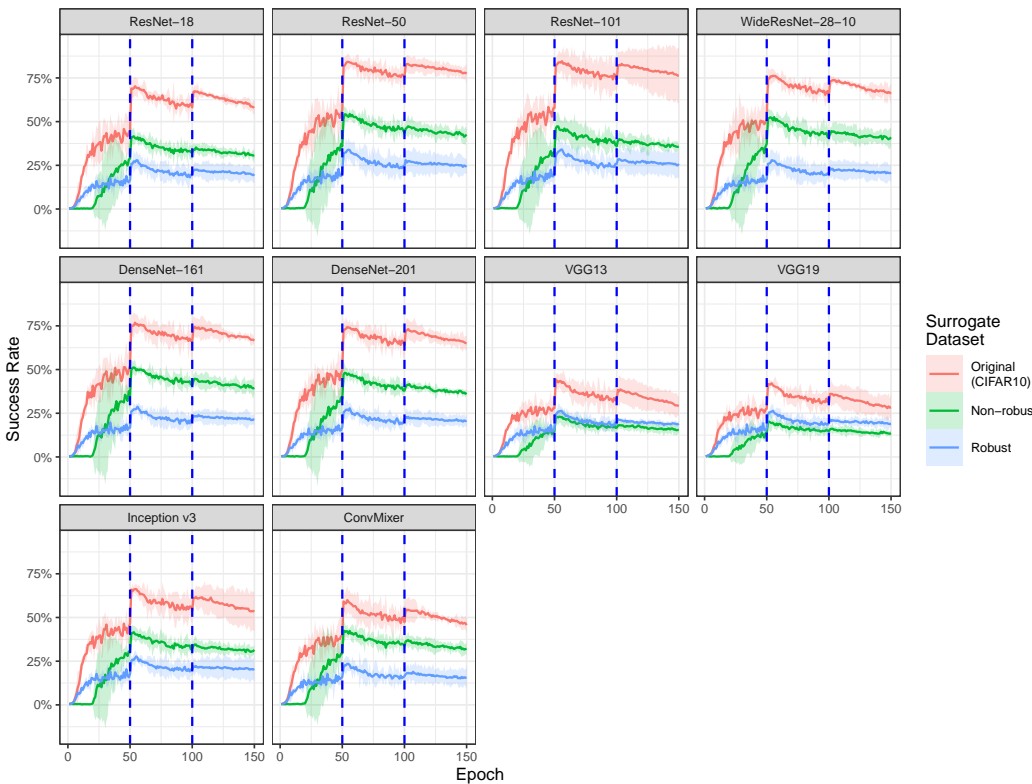

Figure 11: Early stopping improves transferability of surrogate models trained on both robust and non-robust datasets. Success rate evaluated over ten target models (title subfigure) from a ResNet-50 surrogate model trained for a number of epochs (x-axis) on the datasets $D_R$ (blue) and $D_{\text{NR}}$ (green) of Ilyas et al. (2019) modified from CIFAR-10 (red).

# D  TRANSFERABILITY AND TRAINING DYNAMICS

This section contains additional results of Section 4 on the relationship between the training dynamics of the surrogate model and its transferability.

## D.1  CONSISTENCY OF THE PEAK OF TRANSFERABILITY

Figure 12 contains the transferability per target of the surrogate models trained with a single learning rate decay at a varying epoch. The consistency of the peak of transferability across training epochs is valid for all ten targets.

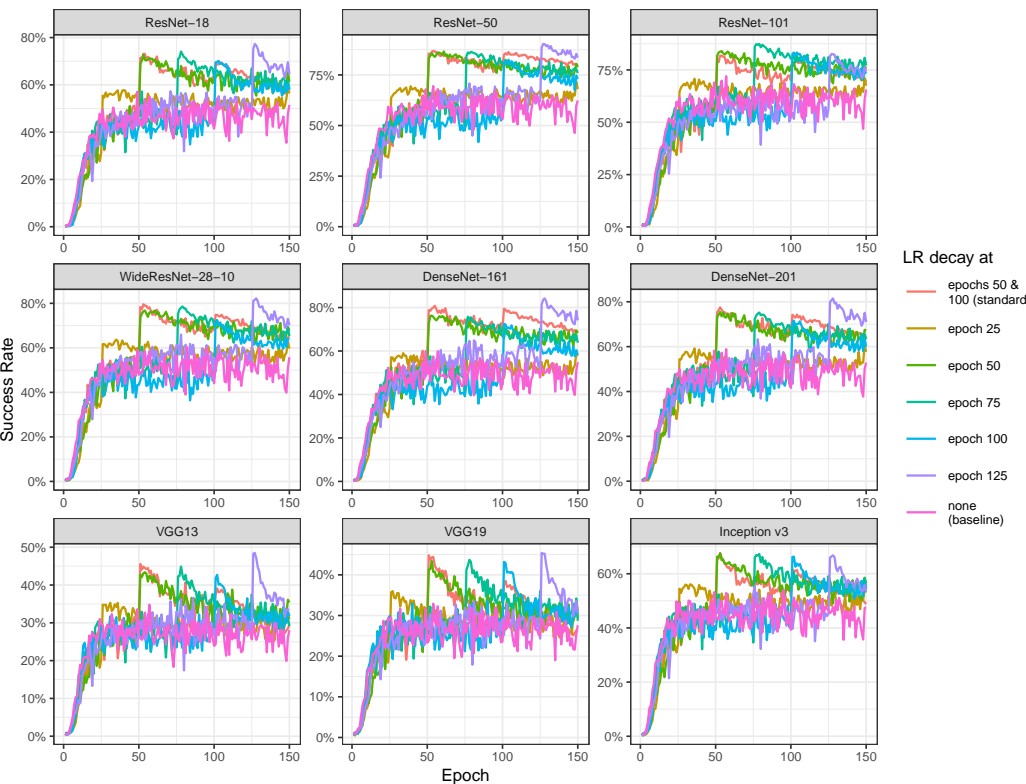

Figure 12: Step learning rate decay benefits transferability at any epochs after initial convergence. Success rate evaluated over nine target models (title subfigure) from a ResNet-50 surrogate model trained for a number of epochs (x-axis) on the CIFAR-10. The LR is divided by 10 a single time during training at an epoch indicated by the colour. Scale not shared between subfigures.

## D.2  CROSSING THE VALLEY BEFORE EXPLORING THE VALLEY

Before the learning rate decays, the exploration tends to behave more like "crossing the valley" than after decay, when it is more likely to "crawl down to the valley", as described in Schneider et al. (2021). Figure 1 illustrates this phenomenon. Schneider et al. (2021) proposes the $\alpha$-quantity, a metric computed at the level of SGD iterations to disentangle whether the iteration understeps or overshoots the minimum along the current step direction. Based on a noise-informed quadratic fit, $\alpha \approx 0$ indicates an appropriate LR that minimizes the loss in the direction of the gradient at this iteration ("going down to the valley"). $\alpha > 0$ indicates that the current LR overshoots this minimum ("crossing the valley"). We compute the $\alpha$-quantity every four SGD iterations during the best five epochs for transferability on CIFAR-10 ("after LR decay", epochs 50–54) and during the five preceding epochs ("before LR decay", epochs 45–49). The one-sided Welch Two Sample t-test has a p-value inferior to $2.2e^{-16}$. We reject the null hypothesis in favor of the alternative hypothesis that the true difference of $\alpha$-quantity in means between the group "before LR decay" and the group

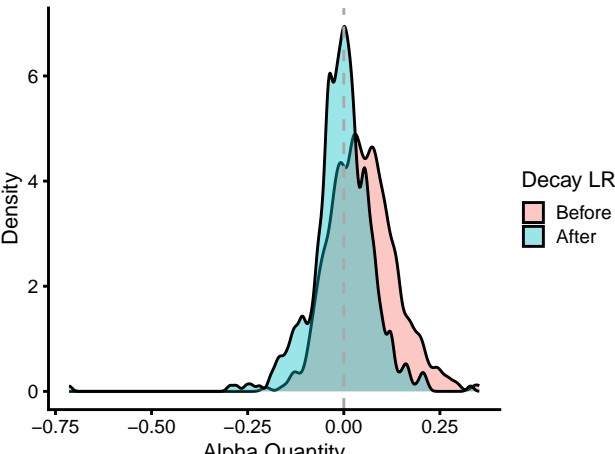

Figure 13: The LR decay corresponds to a transition from a "crossing the valley" phase to a "crawl-ing down to the valley" phase. Density plot of the $\alpha$-quantity values computed each four SGD iterations during the best five epochs for transferability on CIFAR-10 (epochs 50–54, "After" group, blue) and the five preceding epochs (epochs 45–49, "Before" group, red).

"after LR decay" is strictly greater than 0. We also perform a one-sided Welch Two Sample t-test on the 5 epochs before and after the second LR decay (epochs 95–99 vs. epochs 100-105). Its p-value is equal to $0.004387$. Using the Bonferroni correction, we compare the p-values of both individual tests with a significance threshold of 0.5%. We reject the null hypothesis for both LR decays with a significance level of 1%. Figure 13 is the density plot of the $\alpha$-quantities for both groups. Our results suggest that before the LR decay, training is slow due to a "crossing the valley" pattern. The best early stopped surrogate occurs a few training epochs after the LR decay when the SGD starts exploring the bottom of the valley.

# E  TRANSFERABILITY FROM SAM AND ITS VARIANTS

This section presents the following elements:

1. An illustrative schema of SAM (Figure 14),
2. The success rate with respect to the $\rho$ hyperparameter of SAM and its variants, used to tune this hyperparameter for transferability (Section E.1),
3. The natural accuracy of the surrogates trained by SAM and its variants (Section E.2).

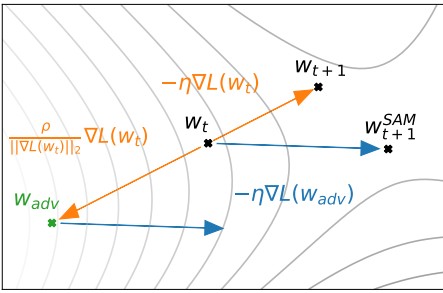

Figure 14: Illustrative schema of a training iteration with SAM. Illustration from Foret et al. (2020).

## E.1  THE SIZE OF FLAT NEIGHBORHOODS: THE CHOICE OF THE $\rho$ HYPERPARAMETER

A stronger regularization induced by SAM with large flat neighborhoods trains a better surrogate model. The size of flat neighborhoods is controlled by the unique hyperparameter of SAM, noted $\rho$. Figure 15 reports the validation success rate used to find the best large $\rho$ for each SAM variants. The selected $\rho$ values are reported for each SAM variant in Table 3. This success rate is computed on a separate set of target models, surrogate models, and a set of examples. This experimental setting is carefully designed to avoid data leakage by optimizing the hyperparameter against specific target models. Otherwise, this could result in model selection, similar to query-based attacks, which are not allowed by our threat model of transfer-based black-box attacks.

LookSAM is an efficient alternative that computes only the additional ascending gradient of SAM once per five training iterations. We faced some convergence issues when applying it with our learning rate schedule (the original authors used a schedule with warmup). To solve this issue, we add some warmup: LookSAM computes both gradients for the first three epochs of training, exactly as SAM. From the fourth epoch, the training resumes to the efficient LookSAM variant. The computational cost reported in Table 1 takes into account this overhead. We also train two additional variants, ASAM, an adaptive variant of SAM, and AGSAM, an adaptive variant of GSAM. We follow the original paper Kwon et al. (2021) to select the hyperparameter $\rho$: the authors recommend multiplying $\rho$ by 10 when switching to an adaptive variant.

Figure 16 reports the test success rate on the same surrogate models, but computed on our test set of target models and using natural examples from the test set. Sections 5 and 6 report results from three other independently trained surrogate models. The transferability improvement of LookSAM with large $\rho$ is tiny compared to LookSAM with the original $\rho$. LookSAM is an efficient variant of SAM that skips 4/5 of the additional ascending gradients of SAM. Our hypothesis is that training with large $\rho$ requires a more refined update strategy.

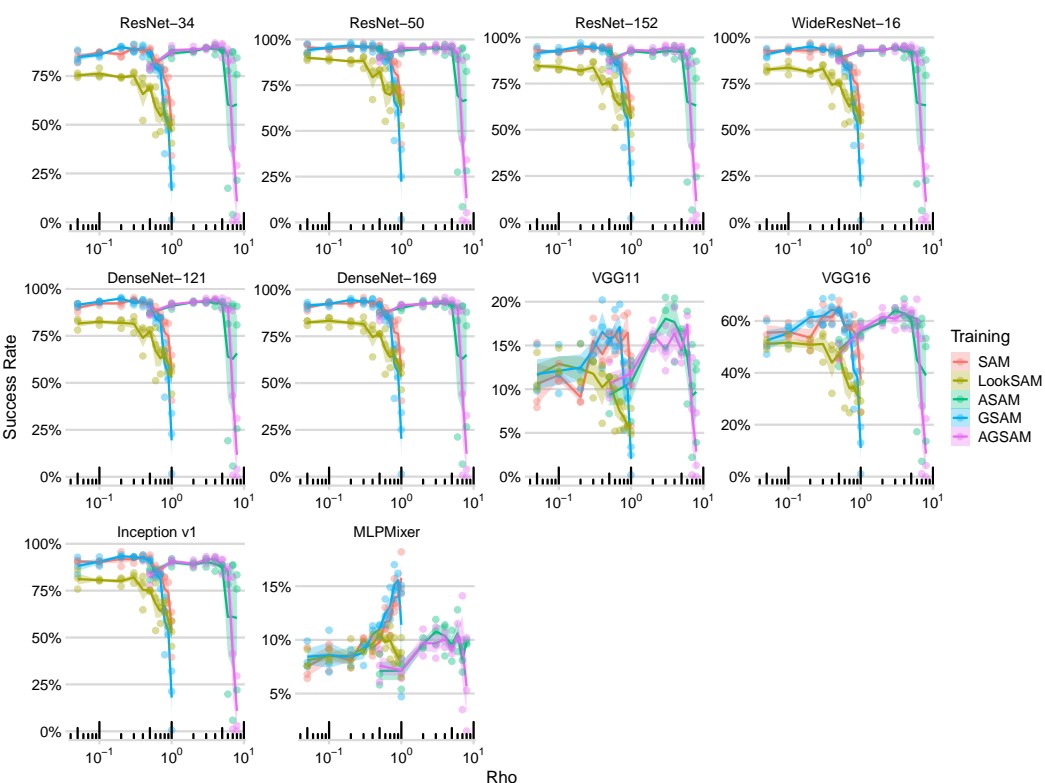

Figure 15: All SAM variants trains better surrogate models with a larger $\rho$ than the original one, used for natural accuracy. *Validation* success rate on ten validation target models (subfigure title) from a ResNet-50 surrogate model trained using SAM or SAM variants (colors) with various $\rho$ hyperparameters (x-axis) on the CIFAR-10 dataset. Adversarial examples are crafted from a disjoint subset of one thousand original examples from the train set. The left most $\rho$ value is the original one: 0.5 for adaptive variants (ASAM, AGSAM), 0.05 for others. Average (line) and $\pm$ one standard deviation (colored area) of three training runs.

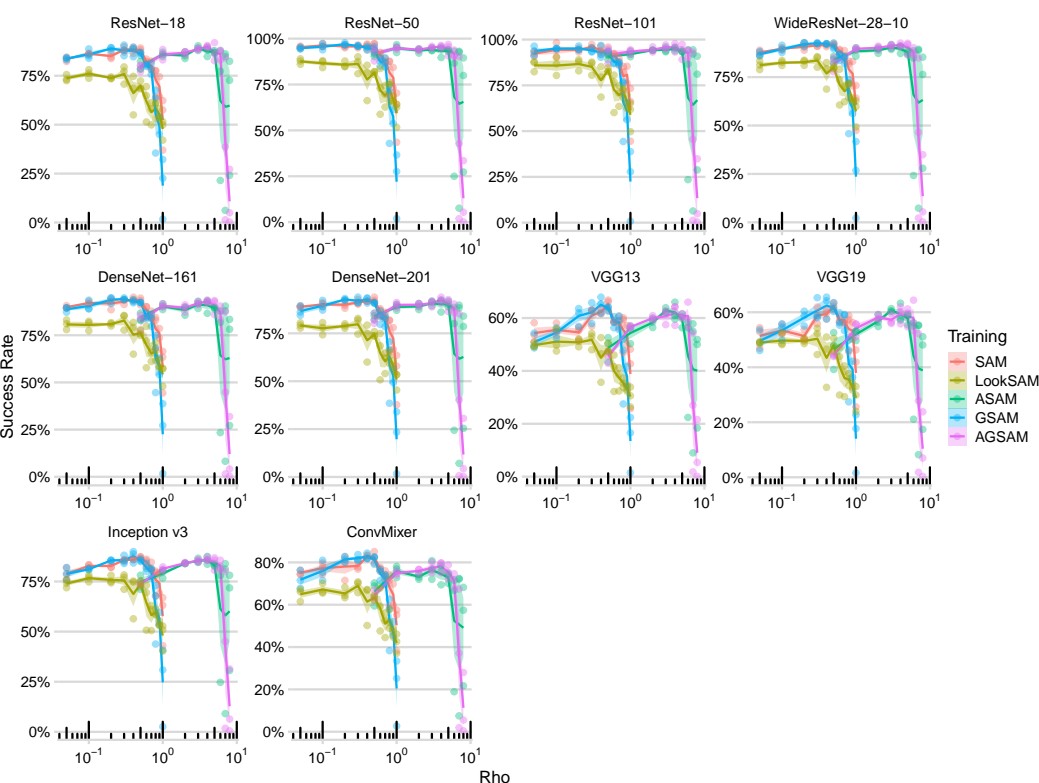

Figure 16: All SAM variants trains better surrogate models with a larger $\rho$ than the original one, used for natural accuracy. *Test* success rate on ten test target models (subfigure title) from a ResNet-50 surrogate model trained using SAM or SAM variants (colors) with various $\rho$ hyperparameters (x-axis) on the CIFAR-10 dataset. Adversarial examples are crafted from a disjoint subset of one thousand original examples from the test set. The left most $\rho$ value is the original one: 0.5 for adaptive variants (ASAM, AGSAM), 0.05 for others. Average (line) and $\pm$ one standard deviation (colored area) of three training runs. The surrogate models used here are the same as in Figure 15. Nevertheless, Sections 5 and 6 report results from three other independently trained surrogate models.

## E.2 NATURAL ACCURACY OF SAM AND ITS VARIANTS

Tables 4 and 5 report the natural test accuracies of the surrogate models studied in Sections 5 and 6. As commented in Section 5, the strong regularization induced by SAM with large flat neighborhoods (high $\rho$) can degrade natural generalization. In particular, on ImageNet, our ResNet-18 and ResNet-50 surrogates trained with l-SAM have a worst natural accuracy compared to SAM and even fully trained SGD. On CIFAR-10, l-SAM has an inferior natural accuracy than SAM, and a similar one to SGD. Therefore, the improvement in transferability from l-SAM, i.e., *the generalization of adversarial examples from this strong regularization, cannot be explained by an improvement in natural generalization*, i.e, a better fit to the data.

Table 4: Accuracy computed on the test set of the surrogates trained by SAM and its variants on ImageNet. In %.

| Arch | Training | Size neighborhood | Accuracy |
|------|----------|-------------------|----------|
| ResNet-18 | SGD (baseline) | None (SGD) | 69.8 |
| ResNet-18 | SAM | Large | 67.9 |
| ResNet-18 | SAM | Original | 70.3 |
| ResNet-18 | GSAM | Large | 68.8 |
| ResNet-18 | GSAM | Original | 70.3 |
| ResNet-18 | ASAM | Large | 68.9 |
| ResNet-18 | ASAM | Original | 70.2 |
| ResNet-18 | AGSAM | Large | 67.8 |
| ResNet-18 | AGSAM | Original | 70.1 |
| ResNet-50 | SGD (baseline) | None (SGD) | 75.7 |
| ResNet-50 | SAM | Large | 74.5 |

Table 5: Accuracy computed on the test set of the surrogates trained by SAM and its variants on CIFAR-10. In %.

| Arch | Training | Size neighborhood | Accuracy |
|------|----------|-------------------|----------|
| ResNet-50 | SGD (baseline) | None (SGD) | 94.5 ±0.4 |
| ResNet-50 | SAM | Large | 94.6 ±0.4 |
| ResNet-50 | SAM | Original | 95.3 ±0.3 |
| ResNet-50 | GSAM | Large | 94.7 ±0.5 |
| ResNet-50 | GSAM | Original | 95.4 ±0.5 |
| ResNet-50 | ASAM | Large | 95.6 ±0.5 |
| ResNet-50 | ASAM | Original | 95.1 ±0.4 |
| ResNet-50 | AGSAM | Large | 95.9 ±0.3 |
| ResNet-50 | AGSAM | Original | 95.3 ±0.6 |

# F   TRANSFERABILITY AND WEIGHT DECAY

We show that in the case of weight decay, a stronger regularization of the surrogate model does not improve transferability. Unlike weight decay, the stronger regularization of SAM is tightly linked to transferability.

We train on CIFAR-10 one surrogate model for various values of the weight decay regularization (5e-3, 1e-3, 5e-4, 1e-4, 5e-5, 1e-5 and 5e-6) and for various capacities of the ResNet architecture (ResNet-18, ResNet-50, ResNet-101). Figure 17 presents the transferability of these surrogates. For the ResNet-50 and ResNet-101 surrogates, the best average success rate simply corresponds to the weight decay used to train the target models. Interestingly, a lighter weight decay regularization trains better ResNet-18 surrogate models. We hypothesize that a ligher regularization allows this smaller architecture to better mimic the complexities of the larger architectures used as targets. Overall, a stronger weight decay regularization does not train better surrogate models, contrary to the SAM regularization.

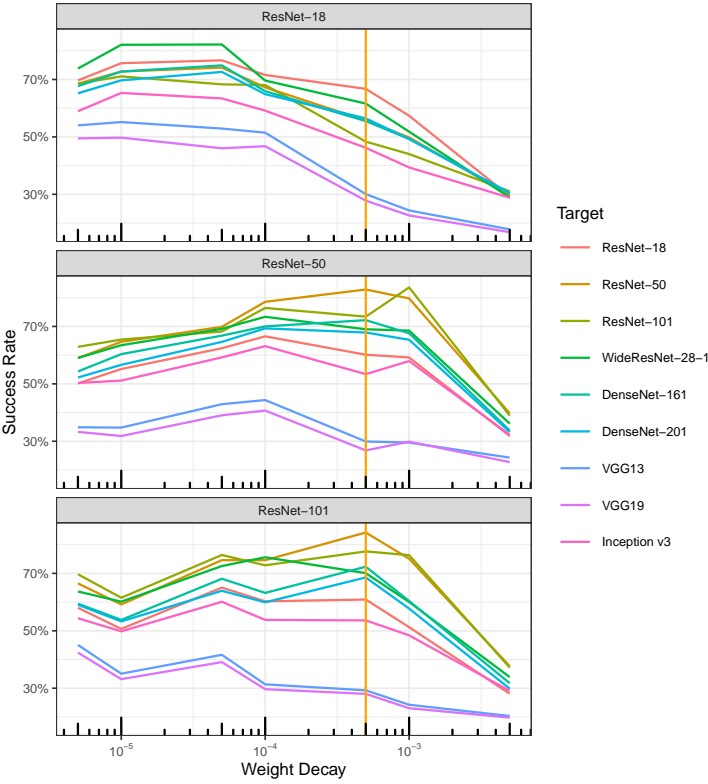

Figure 17: Stronger weight decay regularization does not improve transferability. Success rate from ResNet surrogates (subfigure title) trained with a weight decay (x-axis) evaluated on targets (colors) trained with weight decay indicated by the yellow vertical bar on the CIFAR-10 dataset.

# G   EVALUATION OF l-SAM: IMPROVING TRANSFERABILITY TECHNIQUES WITH SHARPNESS MINIMIZATION

This section extends the evaluation of SAM with large flat neighborhoods of Section 6, performed with $\varepsilon$ equal to $4/255$, to two other perturbation $L_\infty$ norms ($2/255$ and $8/255$) for the competitive techniques, and reports the success rate per target for the complementary techniques.

**Evaluation against competitive techniques.**   Tables 6 and 8 evaluate competitive techniques of l-SAM on CIFAR-10 with, respectively, maximum perturbations $L_\infty$ norm $\varepsilon$ of $2/255$ and $8/255$. The same conclusions made with perturbations of size $4/255$ hold for these two norms: l-SAM clearly improves transferability. l-SAM beats other competitive techniques for the ten targets and both norms. Tables 9, 10, and 11 show, respectively, that l-SAM beats the other techniques in 6 out of 10 targets for $\varepsilon$ equal to $2/255$, and in 5 out of 10 targets for $\varepsilon$ equal to $8/255$.

**Evaluation with complementary techniques.**   Tables 12, 13, and 14 report in detail per target the evaluation of complementary transferability techniques on ImageNet. l-SAM increases the transferability of every eight techniques against every ten targets when combined, except for LGV on 4 targets using $\varepsilon$ equals $2/255$, and LGV on 3 targets with $\varepsilon$ equals $4/255$ or $8/255$. Since LGV collects models with SGD and a high learning rate, a conflict might occur when LGV continues training with SGD from a checkpoint trained with SAM. Future work may explore the adaptation of the LGV model collection to SAM.

Table 6: Success rate on CIFAR-10 of competitive techniques to train a single surrogate model. Adversarial examples evaluated on nine targets with a maximum perturbation $L_\infty$ norm $\varepsilon$ of $2/255$. Bold is best. In %.

|  | Target | | | | | | | | |
| --- | --- | --- | --- | --- | --- | --- | --- | --- | --- |
| Surrogate | RN18 | RN50 | RN101 | DN161 | DN201 | VGG13 | VGG19 | IncV3 | WRN28 |
| Fully Trained SGD | 24.2 | 44.7 | 35.6 | 33.3 | 31.4 | 9.6 | 9.2 | 22.6 | 30.8 |
| Early Stopped SGD | 28.6 | 46.1 | 38.6 | 36.3 | 34.6 | 12.7 | 13.0 | 27.1 | 34.9 |
| SAT | 19.7 | 27.3 | 25.4 | 20.1 | 20.3 | 13.4 | 13.5 | 17.6 | 20.5 |
| l-SAM (ours) | **45.4** | **67.1** | **60.6** | **58.9** | **55.8** | **20.5** | **19.8** | **45.0** | **54.1** |

Table 7: Success rate on CIFAR-10 of competitive techniques to train a single surrogate model. Adversarial examples evaluated on nine targets with a maximum perturbation $L_\infty$ norm $\varepsilon$ of $4/255$. Bold is best. In %.

|  | Target | | | | | | | | |
| --- | --- | --- | --- | --- | --- | --- | --- | --- | --- |
| Surrogate | RN18 | RN50 | RN101 | DN161 | DN201 | VGG13 | VGG19 | IncV3 | WRN28 |
| Fully Trained SGD | 57.9 | 81.2 | 70.6 | 70.8 | 66.1 | 27.8 | 26.3 | 49.4 | 66.5 |
| Early Stopped SGD | 73.3 | 87.8 | 82.1 | 81.4 | 78.3 | 45.5 | 44.3 | 66.8 | 79.5 |
| SAT | 66.3 | 76.2 | 73.6 | 66.9 | 66.1 | 49.8 | 48.5 | 57.9 | 67.8 |
| l-SAM **(ours)** | **89.7** | **97.3** | **95.5** | **95.7** | **94.0** | **63.6** | **60.6** | **87.3** | **93.0** |

Table 8: Success rate on CIFAR-10 of competitive techniques to train a single surrogate model. Adversarial examples evaluated on nine targets with a maximum perturbation $L_\infty$ norm $\varepsilon$ of $8/255$. Bold is best. In %.

| | Target | | | | | | | | |
|---|---|---|---|---|---|---|---|---|---|
| Surrogate | RN18 | RN50 | RN101 | DN161 | DN201 | VGG13 | VGG19 | IncV3 | WRN28 |
| Fully Trained SGD | 88.3 | 97.4 | 92.4 | 93.9 | 91.4 | 64.2 | 60.5 | 79.3 | 91.9 |
| Early Stopped SGD | 97.8 | 99.6 | 98.8 | 98.9 | 98.4 | 89.1 | 87.5 | 95.6 | 98.8 |
| SAT | 97.0 | 98.7 | 98.0 | 97.1 | 96.4 | 90.2 | 89.2 | 93.2 | 97.1 |
| l-SAM (ours) | **99.7** | **100.0** | **100.0** | **100.0** | **99.9** | **96.6** | **95.6** | **99.6** | **99.9** |

Table 9: Success rate on ImageNet of competitive techniques to train a single surrogate model. Adversarial examples evaluated on ten targets with a maximum perturbation $L_\infty$ norm $\varepsilon$ of $2/255$. Bold is best. In %.

| | Target | | | | | | | | | |
|---|---|---|---|---|---|---|---|---|---|---|
| Surrogate | RN50 | RN152 | RNX50 | WRN50 | VGG19 | DN201 | IncV1 | IncV3 | ViT B | SwinS |
| Fully Trained SGD | 18.7 | 9.4 | 10.0 | 9.3 | 7.6 | 5.8 | 4.8 | 5.2 | 1.1 | 1.4 |
| Early Stopped SGD | 23.8 | 10.7 | 10.6 | 10.6 | 8.7 | 6.8 | 5.6 | 6.1 | 1.1 | 1.5 |
| LGV-SWA | 49.3 | 24.8 | 25.0 | 21.7 | 18.5 | 16.8 | 11.6 | 7.9 | 1.4 | 1.5 |
| SAT | 30.0 | 19.2 | 24.4 | 20.6 | 18.4 | 20.2 | **20.0** | **16.6** | **4.9** | **4.4** |
| l-SAM (ours) | **53.3** | **34.3** | **37.5** | **38.3** | **30.7** | **25.0** | 16.6 | 10.8 | 1.7 | 3.8 |

Table 10: Success rate on ImageNet of competitive techniques to train a single surrogate model. Adversarial examples evaluated on ten targets with a maximum perturbation $L_\infty$ norm $\varepsilon$ of $4/255$. Bold is best. In %.

| | Target | | | | | | | | | |
|---|---|---|---|---|---|---|---|---|---|---|
| Surrogate | RN50 | RN152 | RNX50 | WRN50 | VGG19 | DN201 | IncV1 | IncV3 | ViT B | SwinS |
| Fully Trained SGD | 44.5 | 25.2 | 24.8 | 27.1 | 16.2 | 16.4 | 9.8 | 8.0 | 1.8 | 3.3 |
| Early Stopped SGD | 51.5 | 27.4 | 27.7 | 28.0 | 18.4 | 18.7 | 10.8 | 10.4 | 2.2 | 2.7 |
| LGV-SWA | 82.5 | 56.8 | 58.5 | 54.0 | 40.9 | 42.4 | 28.3 | 15.1 | 3.1 | 5.7 |
| SAT | 76.3 | 62.5 | 66.8 | 63.4 | 48.1 | **59.0** | **47.9** | **40.8** | **17.4** | **16.8** |
| l-SAM **(ours)** | **85.7** | **70.3** | **73.3** | **73.2** | **58.2** | 55.6 | 37.9 | 20.5 | 4.0 | 8.2 |

Table 11: Success rate on ImageNet of competitive techniques to train a single surrogate model. Adversarial examples evaluated on ten targets with a maximum perturbation $L_\infty$ norm $\varepsilon$ of $8/255$. Bold is best. In %.

| | Target | | | | | | | | | |
|---|---|---|---|---|---|---|---|---|---|---|
| Surrogate | RN50 | RN152 | RNX50 | WRN50 | VGG19 | DN201 | IncV1 | IncV3 | ViT B | SwinS |
| Fully Trained SGD | 77.5 | 52.9 | 51.1 | 55.0 | 33.4 | 36.9 | 21.1 | 15.2 | 3.7 | 6.7 |
| Early Stopped SGD | 82.0 | 56.8 | 54.6 | 59.2 | 35.9 | 41.1 | 24.8 | 18.3 | 3.6 | 5.9 |
| LGV-SWA | 96.9 | 87.7 | 87.1 | 84.9 | 65.4 | 72.8 | 56.8 | 31.2 | 7.0 | 12.3 |
| SAT | 95.4 | 92.6 | 93.0 | 92.8 | 79.0 | **90.1** | **79.1** | **66.3** | **38.5** | **39.1** |
| l-SAM (ours) | **97.6** | **92.8** | **93.8** | **95.3** | **83.2** | 85.5 | 71.2 | 42.3 | 9.1 | 19.0 |

Table 12: Success rate on ImageNet of three complementary categories of transferability techniques evaluated on ten targets with a maximum perturbation $L_\infty$ norm $\varepsilon$ of $2/255$. Underlined is worse when combined with l-SAM. In %.

| | | | | | Target | | | | | |
|---|---|---|---|---|---|---|---|---|---|---|
| Attack | RN50 | RN152 | RNX50 | WRN50 | VGG19 | DN201 | IncV1 | IncV3 | ViT B | SwinS |
| **Model Augmentation Techniques** | | | | | | | | | | |
| GN | 34.6 | 17.9 | 17.4 | 18.0 | 12.7 | 10.4 | 8.1 | 6.3 | 1.3 | 2.0 |
| GN + l-SAM | 59.7 | 42.2 | 42.8 | 45.4 | 35.4 | 29.1 | 18.3 | 11.0 | 1.8 | 2.4 |
| SGM | 26.9 | 14.9 | 15.2 | 15.8 | 15.5 | 9.7 | 7.4 | 6.6 | 1.6 | 3.6 |
| SGM + l-SAM | 46.3 | 32.0 | 33.8 | 35.5 | 33.4 | 21.8 | 20.3 | 11.9 | 2.7 | 5.6 |
| LGV | 59.8 | 33.0 | 32.9 | 28.4 | 31.1 | 24.2 | 21.3 | 12.5 | 2.4 | 2.6 |
| LGV + l-SAM | 50.7 | 31.3 | 32.9 | 31.4 | 33.5 | 27.9 | 25.0 | 14.3 | 2.1 | 2.5 |
| **Data Augmentation Techniques** | | | | | | | | | | |
| DI | 46.1 | 27.2 | 30.9 | 30.3 | 22.4 | 24.8 | 17.8 | 15.0 | 2.5 | 4.1 |
| DI + l-SAM | 66.6 | 49.5 | 57.1 | 52.3 | 54.1 | 49.3 | 47.5 | 31.8 | 4.4 | 6.9 |
| SI | 26.2 | 14.2 | 14.3 | 13.3 | 10.4 | 11.3 | 8.4 | 7.3 | 0.9 | 1.4 |
| SI + l-SAM | 56.5 | 37.9 | 42.9 | 41.2 | 33.0 | 31.4 | 25.0 | 14.7 | 2.1 | 2.9 |
| VT | 26.5 | 14.4 | 14.1 | 13.6 | 10.8 | 10.1 | 6.1 | 6.3 | 1.3 | 2.2 |
| VT + l-SAM | 61.5 | 43.0 | 47.0 | 47.4 | 39.4 | 35.9 | 24.3 | 13.3 | 2.1 | 4.9 |
| **Attack Optimizers** | | | | | | | | | | |
| MI | 29.8 | 15.9 | 16.4 | 16.2 | 12.6 | 11.5 | 7.7 | 8.0 | 1.9 | 2.7 |
| MI + l-SAM | 58.2 | 41.5 | 45.4 | 44.6 | 39.8 | 35.8 | 28.9 | 17.4 | 2.8 | 5.4 |
| NI | 21.1 | 11.0 | 10.9 | 11.2 | 8.4 | 6.9 | 5.0 | 5.2 | 1.3 | 1.7 |
| NI + l-SAM | 44.1 | 28.5 | 30.7 | 32.0 | 25.9 | 19.6 | 11.9 | 9.1 | 1.3 | 2.4 |

Table 13: Success rate on ImageNet of three complementary categories of transferability techniques evaluated on ten targets with a maximum perturbation $L_\infty$ norm $\varepsilon$ of $4/255$. Underlined is worse when combined with l-SAM. In %.

| | | | | | Target | | | | | |
|---|---|---|---|---|---|---|---|---|---|---|
| Attack | RN50 | RN152 | RNX50 | WRN50 | VGG19 | DN201 | IncV1 | IncV3 | ViT B | SwinS |
| **Model Augmentation Techniques** | | | | | | | | | | |
| GN | 68.0 | 43.1 | 41.3 | 44.1 | 24.8 | 27.2 | 14.3 | 9.9 | 1.9 | 3.8 |
| GN + l-SAM | 89.6 | 76.6 | 79.4 | 79.9 | 65.7 | 60.3 | 42.2 | 22.4 | 3.8 | 7.8 |
| SGM | 62.8 | 40.6 | 41.5 | 43.5 | 31.9 | 28.0 | 19.3 | 13.2 | 4.1 | 7.9 |
| SGM + l-SAM | 83.2 | 68.7 | 71.5 | 73.0 | 67.0 | 56.2 | 48.9 | 26.6 | 6.2 | 13.6 |
| LGV | 93.3 | 78.1 | 75.3 | 73.1 | 64.4 | 61.6 | 49.3 | 28.8 | 5.0 | 6.5 |
| LGV + l-SAM | 88.7 | 74.3 | 75.7 | 75.7 | 70.3 | 61.9 | 56.8 | 31.5 | 4.5 | 7.3 |
| **Data Augmentation Techniques** | | | | | | | | | | |
| DI | 83.1 | 60.5 | 68.1 | 67.3 | 45.4 | 57.9 | 41.4 | 30.7 | 5.7 | 9.9 |
| DI + l-SAM | 95.0 | 89.7 | 90.7 | 91.6 | 85.3 | 87.8 | 87.5 | 64.2 | 14.2 | 19.0 |
| SI | 60.0 | 37.9 | 37.3 | 40.0 | 23.9 | 30.0 | 19.6 | 13.5 | 2.6 | 3.8 |
| SI + l-SAM | 89.2 | 76.6 | 80.1 | 79.1 | 65.2 | 69.8 | 58.0 | 35.8 | 5.0 | 8.5 |
| VT | 58.6 | 35.0 | 35.2 | 38.5 | 23.9 | 24.7 | 14.9 | 11.0 | 2.3 | 4.9 |
| VT + l-SAM | 92.0 | 81.2 | 82.4 | 82.9 | 72.3 | 72.3 | 56.7 | 33.6 | 7.0 | 13.5 |
| **Attack Optimizers** | | | | | | | | | | |
| MI | 56.8 | 37.4 | 37.5 | 38.9 | 27.0 | 29.3 | 18.4 | 14.6 | 3.5 | 4.8 |
| MI + l-SAM | 89.4 | 79.3 | 80.4 | 80.8 | 71.5 | 71.1 | 60.1 | 39.3 | 8.5 | 15.2 |
| NI | 53.7 | 33.1 | 32.9 | 35.1 | 20.5 | 20.8 | 12.2 | 9.4 | 1.8 | 3.9 |
| NI + l-SAM | 83.9 | 67.3 | 69.8 | 71.4 | 56.1 | 52.5 | 35.6 | 17.6 | 3.8 | 7.0 |

Table 14: Success rate on ImageNet of three complementary categories of transferability techniques evaluated on ten targets with a maximum perturbation $L_\infty$ norm $\varepsilon$ of $8/255$. Underlined is worse when combined with l-SAM. In %.

| Attack | Target | | | | | | | | | |
|---|---|---|---|---|---|---|---|---|---|---|
| | RN50 | RN152 | RNX50 | WRN50 | VGG19 | DN201 | IncV1 | IncV3 | ViT B | SwinS |
| **Model Augmentation Techniques** | | | | | | | | | | |
| GN | 92.0 | 73.3 | 69.7 | 74.5 | 45.8 | 50.4 | 29.8 | 19.2 | 3.2 | 7.1 |
| GN + l-SAM | 98.2 | 96.5 | 96.5 | 97.4 | 87.3 | 88.3 | 74.4 | 42.9 | 9.0 | 19.4 |
| SGM | 91.2 | 78.4 | 76.2 | 79.2 | 65.1 | 59.7 | 48.2 | 29.1 | 8.9 | 19.6 |
| SGM + l-SAM | 97.3 | 95.1 | 96.4 | 96.5 | 91.5 | 88.7 | 84.8 | 59.8 | 18.9 | 32.8 |
| LGV | 99.6 | 97.4 | 95.9 | 95.7 | 87.7 | 91.7 | 79.9 | 47.9 | 8.9 | 16.4 |
| LGV + l-SAM | 99.0 | 96.5 | 96.2 | 96.7 | 90.7 | 91.0 | 85.7 | 53.8 | 9.5 | 17.7 |
| **Data Augmentation Techniques** | | | | | | | | | | |
| DI | 96.1 | 90.7 | 91.8 | 91.4 | 74.1 | 88.1 | 72.4 | 55.0 | 14.2 | 20.4 |
| DI + l-SAM | 99.8 | 99.6 | 99.5 | 99.7 | 98.6 | 99.3 | 98.4 | 90.4 | 34.7 | 48.7 |
| SI | 90.4 | 70.0 | 69.9 | 71.9 | 47.8 | 60.2 | 42.6 | 29.3 | 6.5 | 10.3 |
| SI + l-SAM | 98.9 | 97.3 | 97.3 | 98.0 | 89.9 | 94.8 | 90.1 | 67.2 | 15.0 | 23.6 |
| VT | 79.6 | 62.8 | 61.1 | 63.5 | 41.9 | 48.4 | 32.2 | 23.3 | 6.3 | 10.5 |
| VT + l-SAM | 98.0 | 96.7 | 96.1 | 97.3 | 92.9 | 93.1 | 87.1 | 64.2 | 20.0 | 39.2 |
| **Attack Optimizers** | | | | | | | | | | |
| MI | 83.3 | 60.9 | 63.3 | 64.3 | 48.7 | 53.8 | 39.2 | 30.4 | 7.4 | 11.7 |
| MI + l-SAM | 98.5 | 96.3 | 96.7 | 97.1 | 91.9 | 92.7 | 88.1 | 68.6 | 21.3 | 31.5 |
| NI | 86.2 | 65.3 | 65.1 | 70.3 | 43.6 | 47.1 | 28.7 | 19.6 | 4.7 | 8.3 |
| NI + l-SAM | 97.9 | 94.0 | 95.0 | 96.0 | 87.3 | 86.2 | 74.2 | 42.6 | 10.7 | 21.0 |

