# OpenReview forum: "Going Further: Flatness at the Rescue of Early Stopping for Adversarial Example Transferability"
_ICLR.cc/2024/Conference — Submitted to ICLR 2024_

### Official Review · Reviewer_2E6M · 2023-10-29

**Soundness:** 2 fair
**Presentation:** 1 poor
**Contribution:** 2 fair
**Rating:** 3
**Confidence:** 4

**Summary:**

This paper aims to increase the transferability of adversarial attacks, by focusing on optimizing the surrogate model rather than the attack objective.
To enhance attack transferability, the authors employ SAM (sharpness-aware minimization) to train a surrogate model with a flatter loss landscape w.r.t. parameters so that adversarial attack optimization does not fall into local minima.

The authors first examine the relationship between non-robust features (Ilyas et al.) and the improvement in transferability resulting from early stopping in surrogate model training. Previous research suggests that "slightly robust features" transfer more effectively than non-robust features. However, this paper demonstrates that "early stopped non-robust features" transfer better than "fully-trained non-robust features," challenging the existing hypothesis.

Next, they explore the connection between the sharpness of the loss surface with respect to the parameters of a surrogate model and adversarial transferability. Firstly, they demonstrate that the transferability decreases as the learning rate decays during surrogate model training due to an increase in the sharpness of the loss surface. Secondly, they illustrate that using SAM (sharpness-aware minimization), particularly with a larger penalty than the default (denoted as l-SAM in the paper), can enhance transferability. Finally, they show that their method complements existing approaches and further enhances transferability.

**Strengths:**

S1. Their method of employing SAM with a large penalty (i.e., l-SAM) to train a surrogate model improves the transferability.

S2. l-SAM complements existing approaches and further enhances transferability.

**Weaknesses:**

W1. Throughout the paper, there is no mathematical equation, and it lacks the necessary description to understand the paper.
- (W1-1) The existing hypothesis and the authors' claim in Sec. 3 regarding the relation between the non-robust features hypothesis and early stopping are vague and difficult to understand since there is no mathematical formulation.
- (W1-2) SAM should be described with a mathematical equation. SAM hyperparameter $\rho$ is not described in the paper (requires referring to the original paper).
- (W1-3) There are no descriptions of SAM variants (GSAM [Zhuang et al. 2022], ASAM [Kwon et al. 2022], and LookSAM [Liu et al. 2022]) used in their experiments, making their experiments difficult to interpret.

W2.
I do not see any contradiction between the existing hypothesis and the experimental results in  Section 3 regarding the relation between the non-robust features hypothesis and early stopping.
As far as I understand, [Benz et al. 2021] claim robust features transfer better than non-robust features. The authors' experiments show that "early-stopped non-robust features" are more transferable than "fully-trained non-robust features," which is aligned with [Benz et al. 2021] since "early-stopped non-robust features" should be more robust than "fully-trained non-robust features." (Anyway, this discussion requires mathematical formulation to discuss precisely.)

W3. The fact that the loss-surface sharpness of a surrogate model affects the transferability has already been shown by [Gubri et al. 2022], hence reducing the scientific contribution of this paper.
[Gubri et al. 2022] proposed a method called LGV that aims to enhance the loss-surface flatness of the surrogate model, which is the same motivation as this paper.
In fact, in Table 2, the difference between LGV and LGV+l-SAM is minor, making the contribution weak.
The authors should make clear what's the difference between LGV and their approach and their contribution.

-------------------
References

[Benz et al. 2021] Batch Normalization Increases Adversarial Vulnerability and Decreases Adversarial Transferability: A Non-Robust Feature Perspective. ICCV 2021

[Gubri et al. 2022] LGV: Boosting Adversarial Example Transferability from Large Geometric Vicinity. ECCV2022

[Springer et al. 2021] A Little Robustness Goes a Long Way: Leveraging Robust Features for Targeted Transfer Attacks. NeurIPS 2021

**Questions:**

Q1. In Figure 5, the success rate of the transfer attack decreases even when the sharpness of the surrogate model decreases after the 100th epoch. How can it be explained?

Q2. It lacks discussion of attacking adversarially trained models, as discussed by [Wu et al. 2020], [Springer et al. 2021]. Is l-SAM useful for attacking adversarially trained models as well?

Q3. What is the loss-surface sharpness value for other methods, such as SAT [Springer et al. 2021]? I would expect SAT to be sharper than l-SAM based on the paper's claim. If it is, it can strengthen the contribution. If it's not, I would like to know the explanation for why the SAT is less transferable. In other words, does the sharpness metric correlate well with the transferability?

-------------------
References

[Wu et al 2020] SKIP CONNECTIONS MATTER: ON THE TRANSFERABILITY OF ADVERSARIAL EXAMPLES GENERATED
WITH RESNETS. ICLR 2020

[Springer et al. 2021] A Little Robustness Goes a Long Way: Leveraging Robust Features for Targeted Transfer Attacks. NeurIPS 2021

---

> ### Author Response · Authors · 2023-11-23
> **Answer to reviewer 2E6M**
>
> Thanks for your detailed review. We address the questions and weaknesses below.
>
> ### Questions
>
> **Q1.** Thanks for raising this point. We leave as open question why the third phase of training behave differently than the second one after the learning rate decay. We will clarify this point.
>
> **Q2.** Thanks for your suggestion. We will add the evaluation on adversarially trained target model in the next revision.
>
> **Q3**. Thanks for the suggestion. We will add the sharpness metrics for SAT and other surrogate training techniques in Table 1 in the next revision. We will discuss the results accordingly.
>
> ### Weaknesses
>
> **(W1)** Thanks for your suggestions. We will improve the presentation by including formal mathematical formulations of non-robust features, SAM and its variants. Due to lack of space, we moved the description of the five SAM variants to the Appendix. We will do our best to improve their presentation.
>
> **(W2)** We respectfully and kindly disagree: our empirical observations in Section 3 are in contradiction with the conclusion of \[Benz et al. 2021\] that "DNNs first mainly learn Robust Features (RFs) and then Non-Robust Features (NRFs)", and that "to get a substitute model with more Robust Features, a straightforward idea inspired by this finding is to train the substitute model with an early stop" (page 8 of \[Benz et al. 2021\]). We agree with you that early learned NRFs are more transferable than fully learned NRFs (our Figure 2, green line). This conclusion conflicts with the explanation of \[Benz et al. 2021\] that early stopped improves transferability due to an inherent trade-off between RFs and NRFs. As a complement, Figure 3 shows that early stopped representations are best to target both NRFs and RFs targets. Considering transferability as a proxy to measure representation similarity, we do not observe such a trade-off between RFs and NRFs, along epochs. We will clarify our position.
>
> **(W3)** Thanks. We agree that we have the same motivation as \[Gubri et al. 2022\]. Our work complements \[Gubri et al. 2022\] for two reasons.
>
> 1. Contrary to \[Gubri et al. 2022\], we explicitly optimize the sharpness of the surrogate model. The high learning rate used by LGV \[Gubri et al. 2022\] implicitly affects sharpness.
> 2. \[Gubri et al. 2022\] trains an ensemble of surrogates, whereas we train a single DNN. \[Gubri et al. 2022\] observes that a single LGV model transfers poorly, leaving as an open question whether we can improve the transferability of a single representation using flatness. We address this question.
>
> We will improve our current discussion of LGV \[Gubri et al. 2022\] in the related work.

---

### Official Review · Reviewer_5DVg · 2023-10-30

**Soundness:** 3 good
**Presentation:** 3 good
**Contribution:** 3 good
**Rating:** 6
**Confidence:** 4

**Summary:**

This paper aims to enhance the transferability of adversarial examples. Firstly, it provides empirical evidence that challenges the claim made by previous studies, which attribute the effectiveness of early-stopping on transferability to robust or non-robust features. Additionally, this paper establishes a correlation between the peak of transferability during training and both the decay of learning rate and the sharpness of the loss landscape. Based on this observation, the paper proposes the utilization of sharpness-aware minimization (SAM) as the optimizer for training a surrogate attack model, which can effectively improve the success rate of transfer attacks.

**Strengths:**

1. This paper is clearly written and well-organized.
2. The empirical evidence is sufficient to support the claims related to learning rate decay and early stopping. The experiments were conducted using various datasets and model architectures.
3. The proposed method is clear and can be easily incorporated into other existing transfer attack methods.
4. The experiment settings are clearly described, and all hyper-parameters are provided.

**Weaknesses:**

1. Though supported by experimental evidence, there is a lack of theoretical justification for why sharpness has a strong connection to transferability.
2. The evaluation experiments are only conducted on standardly trained models. I wonder how l-SAM works when attacking adversarially trained robust models.
3. As acknowledged in Section 2, the underlying mechanism of SAM is still a popular research topic and remains controversial [1, 2]. In particular, it is still uncertain whether the effectiveness of SAM is caused by sharpness. Therefore, I suggest toning down some claims on the proposed method, such as:

    > the effect of early stopping on transferability is closely related to the dynamics of the exploration of the loss surface
    >
4. In Section 2, I suggest adding more details (e.g. training objective and algorithm) to improve readability, particularly for readers who are not familiar with SAM.
5. In the context of adversarial robustness, leveraging SAM is not a completely new idea. While I'm not critiquing the novelty of this paper, I think the following papers [3,4] should be mentioned in the related work.

    a. [3] proposes leveraging SAM to craft adversarial examples to find the common weaknesses of different models, which can boost the transferability of adversarial attacks.

    b. [4] shows that using SAM with a larger $\rho$ (exactly l-SAM in this paper) in standard training can improve adversarial robustness. Additionally, as shown in [5], a little robustness can improve adversarial transferability, so the effectiveness of using SAM in surrogate training is somewhat expected. Please discuss this viewpoint in your revision.

[1] A modern look at the relationship between sharpness and generalization. ICML

[2] Sharpness Minimization Algorithms Do Not Only Minimize Sharpness To Achieve Better Generalization. NeurIPS

[3] Rethinking Model Ensemble in Transfer-based Adversarial Attacks. arxiv:2303.09105

[4] Sharpness-Aware Minimization Alone can Improve Adversarial Robustness. ICML Workshop

[5] A Little Robustness Goes a Long Way: Leveraging Robust Features for Targeted Transfer Attacks. NeurIPS

**Questions:**

Please see the weaknesses above.

---

> ### Author Response · Authors · 2023-11-23
> **Answer to Reviewer 5DVg**
>
> Thanks for your kind words and for your insightful suggestions to improve the paper. We address the weaknesses below:
>
> 1. We agree that our work does not contain theoretical justification of the relation between sharpness and transferability. Nevertheless, we think that our extensive experimental work is a step-forward in the field to better understand why some representations are better surrogates than others.
> 2. Thanks for your suggestion. We will add the evaluation on adversarially trained target model in the next revision.
> 3. Thanks for the suggestion. We agree with your view: we also mentioned in the related work that the topic of sharpness is controversial. We will better reflect both sides of the controversy in the rest of the paper, as you suggested.
> 4. Thank you for recommending improving the presentation of SAM. For the sake of space, we moved the illustration of SAM to Appendix E. We will improve the presentation of SAM in the main paper.
> 5. Thanks for the relevant references. We already evaluate the technique of \[5\], called SAT (slight adversarial training) in Section 6, and discuss its relation with sharpness in the related work. We will add and discuss \[3\] and \[4\] in the related work, as you suggested.

---

### Official Review · Reviewer_hxCY · 2023-11-04

**Soundness:** 1 poor
**Presentation:** 3 good
**Contribution:** 2 fair
**Rating:** 3
**Confidence:** 5

**Summary:**

This paper examines the impact of early stopping on the transferability of adversarial examples in deep neural networks, revealing that the benefits are due to the effect on the learning dynamics, particularly the exploration of the loss landscape, rather than the learning of robust features, as what the community always believe. It demonstrates that transferability is linked to times when the learning rate decays and loss sharpness decreases. The sharpness-aware optimizer SAM is used, which enhances transferability beyond early stopping. The study also finds a strong correlation between the regularization effects of SAM and increased transferability, positioning sharpness-aware optimization as an effective approach for creating transferable adversarial examples.

**Strengths:**

The strengths of this paper can be summarized as follows:
1. The paper empirically contests the widely accepted notion that early stopping in training DNNs leads to more robust feature learning and thus better transferability of adversarial examples. This is good to the community.
2. This paper introduces an evaluation of various flat-minima optimizers, showing how these can significantly enhance the transferability of adversarial examples by minimizing loss sharpness.
3. The study's focus on the SAM, particularly its large-neighborhood variant (l-SAM), provides evidence of its effectiveness in avoiding overly specific representations, thereby improving transferability.
4. The paper conducts a comparative analysis of the proposed methods against other training procedures, illustrating their effectiveness and complementarity in improving transferability.

**Weaknesses:**

The weaknesses of this work is listed below:

1. First of all, the teaser figure (Fig. 1) is very confusing. I think the illustration is not clear enough. I cannot extract any information of the "transferrability" other than from the texts. I would recommend you to make the training curve in a color system that brighter colors represent high transferability, right now it seems SGD has the best transferability and the early stopping one has the worst. Also, why is this  statistic-based or just sketch map. If it is the latter case, I would recommend the authors to use the former to make this more convincing.
2. I think both the notion used as well as the literature review of the sharpness for neural network in this work is very limited. First of all, the concept of sharpness has different meanings or evaluation method. Below are a lot of literature that this work missed:

    > [1] Low-Pass Filtering SGD for Recovering Flat Optima in the Deep Learning Optimization Landscape, AISTATS 2022

    > [2] A modern look at the relationship between sharpness and generalization, ICML 2023

    > [3] On large-batch training for deep learning: Generalization gap and sharp minima, ICLR 2017

    > [4]  Fantastic generalization measures and where to find them, ICLR 2020

    > [5] Rethinking parameter counting in deep models: Effective dimensionality revisited (The Hessian-based metrics).
In fact, sharpness even has its own meaning in the context of adversarial training:

    > [6] Evaluating and understanding the robustness of adversarial logit pairing.
To be frank, I do not believe the SAM-based sharpness is a very good metric for sharpness, and the authors need to prove it is meaningful in the context of adversarial training.

3. If early stopping improves adversarial example's transferability, I think Figure 2 can be removed or moved to Appendix. Also, please improve the presentation by referring to the figure when making arguments. For example, on page 4 paragraph "Early stopping indeed increases transferability.", there is no figure referred and it is not friendly to readers.

4. For section 4, I have a general question: what about there is no abrupt learning rate decay used during training, for example the cosine learning rate schedule is very popular in different training settings. Based on this work, does the sharpness always drops during training? This seems a bit odd and contradicts to existing finds saying that better robustness prefers better flatness, see [7] below.

    > [7] Relating Adversarially Robust Generalization to Flat Minima/

5. I think the most significant problem of the evaluation is that, it lacks a serious evaluation on the flatness. For example, the literature [1] provides a list of methods to evaluate the flatness following different metrics. The authors claim better flatness improves tranferability, but forget to measure the flatness. This is not convincing.

**Questions:**

I do not have additional questions. Please refer to my comments in the "Weaknesses" column.

---

> ### Author Response · Authors · 2023-11-23
> **Answer to Reviewer hxCY**
>
> Thanks for your detailed review. We comment on the listed weaknesses below:
>
> 1.  Thanks for your suggestion. We will use the suggested color maps to clarify the relation with transferability. Figure 1 is a sketch illustration. This figure represents a schematic basin of attraction, inspired by Kaddour et al. (2022). We will clarify in the legend that this is *not* a loss plot.
> 2. Thanks for the references. We mentioned in our related work that the relationship between sharpness and natural generalization is subject to scientific controversy. We tried to reflect on both sides of the topic. Nevertheless, will improve our coverage of this extensive topic. Thanks for your suggestion.
>    We would like to respectfully correct one point: our surrogate models are **not** trained with adversarial training. We train our surrogate model using SGD or SAM, then we craft adversarial examples against them that we feed to other target models to evaluate transferability. Our work does not consider sharpness in the context of adversarial training.
> 3. We think that there might be a misunderstanding here. The Figure used in the paragraph "Early stopping indeed increases transferability." (page 4) is already in Appendix B, as you suggested. The point of Figure 2 is not to show that early stopping improves transferability overall. Figure 2 provides evidence against the hypothesis currently accepted by the literature that early stopping learns more robust features than non-robust features. This Figure is key to supporting our strength 1 that you recognized as the most valuable of the paper. We will improve the clarity of the captions of Figures 2 and 3.
> 4. The paper \[7\] suggested by the reviewer studies flatness in the context of adversarial training. Our surrogates are **not** trained with adversarial training, but with standard training (SGD or SAM). Therefore, it could be expected that our flatter model are not more robust.
> 5. Thanks for your suggestion. Figure 5 uses two Hessian-based metrics to measure sharpness: the top-1 eigenvalue and the trace. Most importantly, we observe the effect of sharpness on transferability by evaluating optimizers that are known to decrease sharpness: SAM and five of its variants, that minimizes sharpness explicitly and SWA that does it implicitly. We will add other metrics of sharpness in a new revision of the paper.

---

### Official Review · Reviewer_4sAy · 2023-11-08

**Soundness:** 3 good
**Presentation:** 4 excellent
**Contribution:** 3 good
**Rating:** 5
**Confidence:** 4

**Summary:**

The paper proposes a hypothesis about the properties of the surrogate models used to produce adversarial samples in black-box attacks for improved success rate of the attack. The existing view is that there exist two types of features that model learns sequentially. So the early stopping improves transferability of attacks because the features learned first allow for more transferable attacks. The empirical evidence in the paper shows that both types of features show similar behavior with respect to attack success and the early stopping. The proposed conjecture is that transferability success depends on the learning schedule - it is shown empirically that around the LR decay epochs transferability spikes. This is further empirically connected with flatness wrt model parameters and then an approach to train flatter models (with SAM) as surrogate models is evaluated.

**Strengths:**

The paper proposes an interesting view on the reasons for transferability success. Large amount of experiments on different architectures is provided.
The paper is easy to follow and read.

I find the result that transferability improves with very strong flatness regularization, which hurts natural performance, one of the most interesting in the paper.

**Weaknesses:**

The main issue of the paper is the conceptual mismatch: empirical evidence shows that early stopping helps transferability of the attacks, also that transferability spikes when learning rate decays. Nevertheless, the statement "when LR decays, the sharpness of the parameter space drops" (page 5) is not valid overall. Learning rate does not take part in computation of the sharpness and does not affect it. Smaller learning rate is conjectured to drive models to "sharper" places, but up till now it is a conjecture. And therefore connection from LR experiments to sharpness experiments are not justified. Moreover it is stated that the success of transferability is related to the exploration of the loss surface (page 5), which is also not a precise statement - even with a large learning rate the trajectory of training can be very limited.

Several times the concept of basins of attraction is mentioned (section 5), but it is not precisely defined and it is not explained why this is even useful for the transferability of the attacks. Analogously, several times a very vague term "initial convergence" of the training is used. Please define it properly if it is critical for understanding the explanations, or do not use it.

In the works on adversarial attacks it is very important to describe precisely the attack mode assumed. As I understood, the knowledge of the attacker in the paper is assumed to be almost absolute - so they can train a surrogate model with exactly same architecture, dataset and even training setup. This significantly weakens the challenge of attacking. Moreover, only one type of attack is checked, which might not be enough to make a general conclusion - it might be that exactly BIM attack is affected by the flatness, but PGD for example is not. I would rather suggest to reduce the amount of target architectures checked and the ways to induce the flatness, but add at least one more attack type, that is significantly different from BIM.

The discussion about robust features and non-robust ones is very convoluted. According to the definition provided it should be very hard to create adversarial attacks on the RF, but if they are learned first then how early stopping can improve transferability? Or it makes it harder to create attacks, but they are more universal? The conclusion on page 4, from the robust and non-robust surrogate models training seems to be inverted: early learned NRFs would have low transferability (since it grows with training), but early learned RFs might be this way. Moreover, the experiment with robust and non-robust surrogate model does not seem to prove that the hypothesis about the sequence of learning process (model first learns RFs and then NRFs) is wrong. We still see that there is a significant difference in the success rate between robust and non-robust features. The conclusion that can be made is rather that early stopping effect on transferability is not connected with robustness of features, but not that the models do not learn some features earlier than others and this does not affect transferability.

The conclusion made is that flatness of loss surface with respect to the parameters is defining in the transferability of adversarial attacks. While empirical evidence demonstrates validity of such conclusion, I would suggest to be very careful with distinguishing flatness with respect to parameters and optimization with respect to the input that is performed to generate adversarial attacks. First flatness does not necessarily connect with the second, therefore such conclusion may sound misleading.

Finally, measuring only Hessian eigenvalues and trace is not the most precise way to measure sharpness (see for example Petzka, Henning, et al. "Relative flatness and generalization." Advances in neural information processing systems 34 (2021)).

**Questions:**

1 - What is the precise statement that is made by the paper? If it is that transferability of attacks requires very flat in the parameters models, then how learning rate and exploration of the loss surface connects to this statement? What exactly shows that better transferability in the beginning is not affected by the difference in features learned by the network?

2 - What is the attack model that is assumed in the paper?

**Details Of Ethics Concerns:**

The paper proposes an improved way to create surrogate models for black-box adversarial attacks. An ethics statement is missing.

---

> ### Author Response · Authors · 2023-11-23
> **Answer to Reviewer 4sAy**
>
> Thanks for your detailed comments.
>
> ### Question 1
>
> Thanks for pointing out a potential ambiguity. Our central claim is that flat regions of the loss landscape contain better surrogate models. In Section 4, we study the impact of the learning rate on both sharpness and transferability, to provide a more fine-grained analysis of the transferability of early stopping: the transferability along epochs does not evolve smoothly. We uncover the peaks of transferability that happen when the learning decays. To these peaks of transferability correspond peaks of sharpness, suggesting a relationship. We confirm this relationship in Sections 5 and 6. We show that optimizers known to decrease sharpness (SWA, SAM and 5 SAM variants) train better surrogate models. In particular, the large flat neighborhoods found by SAM and its variants, improve transferability further.
>
> Section 3 challenges the accepted hypothesis that early stopping improves transferability due to a trade-off between robust and non-robust features along epochs. If such a trade-off had applied, we would have observed an “X-shaped” transferability
> curves in Figure 2: increasing transferability from non-robust features and strictly decreasing transferability from robust features. Instead, the transferabilities of both robust and non-robust features evolve similarly along epochs. Considering transferability as a proxy to measure representation similarity, we do not observe such a trade-off between robust and non-robust features, since early stopping is best to transfer from and to both robust and non-robust features.
>
> ### Question 2
>
> Thanks for your question. The threat model used here is standard in the literature on the transferability of adversarial examples. As in most of the related work, we assume that the attacker can train a model on the same training set, but not using the same architecture, nor the same random seed. This is reflected in our experimental settings, since we report the transferability from one architecture to another (e.g., from ResNet-50 to ViT-B-16), trained with different random initialization and random batches. We will write explicitly the threat model considered, and in particular the assumption about the attacker's knowledge.
>
> ### Ethics statement
>
> We will add an ethics statement. Thanks for pointing this.

---

> > ### Comment · Reviewer_4sAy · 2023-11-23
> >
> > I thank the authors for replies, but unfortunately the weaknesses I listed still hold. I encourage authors to revise the paper to be more precise in the statements and connections.

---

### Author Response · Authors · 2023-11-23
**General reply**

We would like to thank all reviewers for their detailed and insightful comments. We provide below the answers to each reviewer to address the questions and offer some clarifications.

---

### Public Comment · ~Yechao_Zhang1 · 2023-12-08
**Related Concurrent Work**

Dear authors and reviewers,

I would like to draw your attention to existing work that closely examines similar methods and ideas discussed in this submission. Specifically, the work of [Zhang et al. (2024)] also explores the use of SAM (sharpness-aware minimization) to train a surrogate model and enhance attack transferability. They also provide a comprehensive overview of adversarial transferability and surrogate model training. I believe it would be valuable to acknowledge and engage with this prior work. Notably, Zhang et al.'s observations and conclusions address some of the questions and concerns raised by the reviewers regarding this submission. Here are some key conclusions from [Zhang et al. (2024)]:



- **A flatter loss landscape w.r.t. input space is important**. Optimizing adversarial examples involves finding local maxima in the input space, and a flatter (smoother) loss landscape in the input space, rather than the parameter space emphasized in this paper, is beneficial for better transferability. This is also supported by the theoretical framework presented in [Yang et al. (2021)].


- **The relationship between parameter and input loss landscapes.**  [Dherin et al. 2022] proved that the gradient regularizing pressure on the parameter space (minimizing  $ \| \nabla_\theta f_\theta(x) \| $) can transfer to the input space
(minimizing $\|\nabla_x f_\theta(x)\|_F$).
 [Zhang et al. (2024)] further empirically showed that gradient regularizations in the parameter space, which lead to a flatter loss landscape in the parameter space (measured by the Hessian), also result in a flatter loss landscape in the input space. SAM, which also aims to optimize the flatness of the parameter space, achieves a flatter input loss landscape. *However, the flatness achieved in the input space by SAM is smaller compared to direct input space gradient regularization.*

- **A flatter parameter loss landscape is not as important as a flatter input loss landscape**. [Zhang et al. 2024] investigates methods that explicitly optimize the flatness of parameter loss landscape (ER:  $L_{e r}(\theta)=L(\theta)+\frac{\lambda_{e r}}{2}\left\|\nabla_\theta L(\theta)\right\|^2$) and input loss landscape (IR: $L_{i r}=\frac{1}{\|\mathbf{S}\|} \sum_{i=1}^{\|\mathbf{S}\|}\left[\ell\left(f\left(x_i\right)\right)+\lambda_{i r}\left\|\nabla_x \ell\left(f\left(x_i\right)\right)\right\|\right]$), respectively. SAT [Springer et al. 2021] and IR both result in flatter input loss landscapes, while ER achieves a flatter parameter loss landscape. However, SAT and IR generally outperform ER.
- **SAM yields significant improvement on input gradient alignment.** [Zhang et al. 2024] empirically found that models trained with SAM consistently exhibit better input gradient alignment compared to models trained with SGD. In other words, $cos\langle\nabla_x \ell(f_{\theta_{sam}}(x), y), \nabla_x \ell(g(x), y)\rangle$ is always greater than $cos\langle\nabla_x \ell(f_{\theta_{sgd}}(x), y), \nabla_x \ell(g(x), y)\rangle$,  regardless of the architecture of $g$ and how $g$ is trained (e.g., augmentation, regularization). This property allows adversarial examples crafted against models trained with SAM to be more generalized than those crafted against models trained with SGD.

In conclusion, Zhang et al. argue that the effectiveness of SAM in enhancing attack transferability can be attributed, to its ability to produce a slightly flatter input loss landscape and significantly improve input gradient alignment.

References

[Zhang et al. 2024] Why Does Little Robustness Help? A Further Step Towards Understanding Adversarial Transferability. IEEE S&P 2024

[Yang et al. 2021] TRS: Transferability Reduced Ensemble via Promoting Gradient Diversity and Model Smoothness. NeurIPS 2021

[Dherin et al. 2022] Why Neural Networks Find Simple Solutions: the Many Regularizers of Geometric Complexity.  NeurIPS 2022

[Springer et al. 2021] A Little Robustness Goes a Long Way: Leveraging Robust Features for Targeted Transfer Attacks. NeurIPS 2021

---

### Meta-Review · Area_Chair_WDzJ · 2023-12-03

**Metareview:**

This paper aims to enhance the transferability of black-box adversarial attacks/adversarial examples by training a better surrogate model from which the adversarial examples are obtained. Previous research found that early stopping of the surrogate model improves transferability. The explanation was from the perspective of robust vs non-robust features. This paper challenges this assumption and shows that the key reason for the improved transferability is the learning rate schedule: transferability spikes when LR decays, which simultaneously has a drop in sharpness. The paper thus uses sharpness-aware minimization to improve the transferability.

**Justification For Why Not Higher Score:**

This paper actually proposes an interesting idea. However, the paper can benefit from another round of revisions. Here are some good suggestions from the reviewers.
- The empirical evidence is not strong enough to connect learning rate with sharpness.
- The reviewers find definitions of some technical terms lacking, such as “basins of attraction”, “initial convergence”, “non-robust features”, “SAM”, etc.
- Limited scope due to absolute knowledge of the attackers and study on one attack.
- The discussions on robust/non-robust features can be improved.
- Better illustration of transferability in the teaser figure. More convincing teaser figure using empirical results instead of a sketch map.
- Different ways of evaluating sharpness/flatness can be used.
- What if there is no abrupt learning rate during training?
- More details on the training methods, SAM, and SAM variants.
- Mentioning other related works, e.g., those on adversarial training and sharpness. The LGV paper.

**Justification For Why Not Lower Score:**

N/A

---

### Decision · Program_Chairs · 2024-01-16

Reject